# Histidine is selectively required for the growth of Myc-dependent dedifferentiation tumours in the *Drosophila* CNS

Francesca Froldi[1,2,†], Panayotis Pachnis[3,†], Milán Szuperák[1,4], Olivia Costas[1,2], Tharindu Fernando[3], Alex P Gould[3] & Louise Y Cheng[1,2,5,*] (iD)

## Abstract

Rewired metabolism of glutamine in cancer has been well documented, but less is known about other amino acids such as histidine. Here, we use *Drosophila* cancer models to show that decreasing the concentration of histidine in the diet strongly inhibits the growth of mutant clones induced by loss of Nerfin-1 or gain of Notch activity. In contrast, changes in dietary histidine have much less effect on the growth of wildtype neural stem cells and *Prospero* neural tumours. The reliance of tumours on dietary histidine and also on histidine decarboxylase (Hdc) depends upon their growth requirement for Myc. We demonstrate that Myc overexpression in *nerfin-1* tumours is sufficient to switch their mode of growth from histidine/Hdc sensitive to resistant. This study suggests that perturbations in histidine metabolism selectively target neural tumours that grow via a dedifferentiation process involving large cell size increases driven by Myc.

**Keywords** dedifferentiation; *Drosophila*; histidine; metabolism; neuroblast
**Subject Categories** Cancer; Neuroscience
**The EMBO Journal (2019) 38: e99895**

## Introduction

One hallmark of tumours is that they display a high demand for certain nutrients, at least in part reflecting their fast growth and proliferation (Cantor & Sabatini, 2012). In aerobic conditions, many types of tumours uptake and metabolise glucose to lactate more actively than other cell types in healthy adult tissues (Jones & Thompson, 2009). However, high glycolytic activity is not universally observed across all tumour types and, more generally, there is a high degree of diversity across cancer cell types in the ways in which metabolism can be rewired. With respect to amino acid metabolism, c-MYC induces fibroblasts to become dependent upon glutamine for survival (Yuneva *et al*, 2007); cysteine deprivation triggers programmed necrosis in von Hippel–Lindau (VHL)-deficient clear-cell renal cancer cells but not in VHL-restored counterparts (Tang *et al*, 2016); and mutation of p53 influences how tumour growth is affected by serine metabolism (Maddocks *et al*, 2013). Hence, the available evidence from these and other studies suggests that amino acid requirements for tumour growth may be driven not only by overall proliferative demands but also by tumour type. A detailed understanding of how some tumours but not others are sensitised to particular amino acids will be essential to inform new cancer therapies based on targeting amino acid metabolism.

The neural stem cells of the *Drosophila* CNS have proved a useful model for the study of cell proliferation and tumorigenesis. *Drosophila* neural stem cells, called neuroblasts (NBs), undergo multiple rounds of asymmetric divisions to self-renew and to give rise to the neurons and glia cells which make up the adult central nervous system (Doe, 2008; Egger *et al*, 2008; Homem & Knoblich, 2012). The majority of NBs (including all the NBs of the ventral nerve cord, VNC) are termed type I NBs and these self-renew and give rise to a smaller daughter cell, the ganglion mother cell (GMC), which divides once to generate two postmitotic neurons or glia (Fig 1A). In the central brain, type II NBs generate larger lineages than their type I counterparts via the production of intermediate neural progenitors (INPs), which in turn divide asymmetrically to generate multiple GMCs and neuronal progeny. Asymmetric cell divisions of the NB ensure that basal cell fate determinants that govern differentiation, such as the transcription factor Prospero (Spana & Doe, 1995; Choksi *et al*, 2006), are exclusively inherited by the GMC (Fig 1A). In the absence of basal cell fate determinants, GMCs fail to undergo terminal differentiation, forming additional NB-like cells that drive inappropriate expansion of the neural stem cell pool, ultimately leading to tumour formation. Similarly, in type II NBs, inactivation of differentiation factors, such as SWI/SNF complex

1   Peter MacCallum Cancer Centre, Parkville, Vic., Australia
2   Sir Peter MacCallum Department of Oncology, University of Melbourne, Parkville, Vic., Australia
3   The Francis Crick Institute, London, UK
4   Department of Psychiatry, Perelman School of Medicine at the University of Pennsylvania, Philadelphia, PA, USA
5   The Department of Anatomy and Neuroscience, University of Melbourne, Parkville, Vic., Australia
    *Corresponding author. Tel: +61 385 561291; E-mail: louise.cheng@petermac.org
    † These authors contributed equally to this work

components or constitutive activation of Notch signalling in the INPs, results in reversion of INPs towards an NB-like progenitor state (Song & Lu, 2011; Eroglu *et al*, 2014).

Recently, it has been shown that tumours can also arise via the dedifferentiation of postmitotic neurons (Carney *et al*, 2013; Froldi *et al*, 2015; Southall *et al*, 2014, Xu *et al*, 2017). This is because neurons require transcription factors such as longitudinal lacking (Lola) and nervous finger-1 (Nerfin-1) in order to maintain their differentiated state. Thus, in the absence of Lola or Nerfin-1, neurons revert to highly proliferative NB-like cells that fail to undergo appropriate differentiation and display several hallmarks of cancer. A reversal of differentiation, or dedifferentiation, occurs in cancers such as glioblastoma multiforme, the most common brain malignancy; whereby, cancerous cells arise from dedifferentiated neurons and astrocytes (Friedmann-Morvinski *et al*, 2012). Therefore, malignant tumours with unrestricted progenitor self-renewal can arise via defective progenitors (GMCs and INPs) or defective neurons. Using a combination of genetic manipulations and a holidic chemically defined diet, we compare the requirements for dietary histidine and its metabolism in different types of *Drosophila* mutant clones. Surprisingly, we find that growth sensitivity of clones to histidine perturbations is closely linked to whether or not their mode of growth from dedifferentiated neurons or INPs is dependent upon Myc.

## Results and Discussion

### Histidine deprivation inhibits the growth of *nerfin-1* but not *pros* clones or wildtype stem cells

Mutations in the zinc finger transcription factor Nerfin-1, which is predominantly expressed in postmitotic neural lineages, provide a useful model for neural dedifferentiation-derived clones in the CNS (Froldi *et al*, 2015; Xu *et al*, 2017). In *nerfin-1* mutant clones,

neurons switch off the expression of differentiation genes in favour of stem cell markers, resulting in tumour-like lineages which exhibit unlimited proliferative potential, that fail to differentiate and are metastatic when transplanted into naive adult hosts (Froldi *et al*, 2015 and Fig 1A). To begin evaluating the effects of diet on clonal growth, *nerfin-1* MARCM clones were induced at 48 h after larval hatching (ALH), and upon adult hatching, clone-bearing flies were fed for 3 days on either a standard *Drosophila* medium (Fed) or nutrient restriction medium (NR; agar/PBS). A ~40% reduction in *nerfin-1* clone volume was observed in NR animals, indicating that clonal growth requires adult dietary nutrients (Fig EV1A). To test systematically the clonal growth response to the depletion of individual essential amino acids (EAAs), which cannot be synthesised *de novo* and must therefore be derived from the diet, we developed a holidic chemically defined diet that supports both larval development and adult survival (CDD, see Materials and Methods). The composition of this CDD differs from that of a previously published *Drosophila* holidic diet (Piper *et al*, 2014). Animals carrying *nerfin-1* clones were raised on standard *Drosophila* medium during development and then subjected as adults to dietary withdrawal of individual EAAs from CDD (Fig EV1B). This EAA deprivation regime effectively depletes adults of most of the internal EAA stores accumulated during development, but it is not severe enough to block basal protein synthesis or to decrease medium-term survival. Withdrawal of most of the EAAs (except for valine) resulted in a reduction in clone volume (measured per CNS) after 9 days of feeding (Fig EV1C). To distinguish general versus specific effects of EAAs on proliferation, we measured the effect of EAA withdrawal on proliferation in the adult ovaries. Most of the EAAs (e.g. leucine and data not shown) were required for mutant clonal growth as well as ovarian stem cell proliferation (Figs EV1C, and 1D and E); in contrast, histidine was selectively required for the growth of *nerfin-1* mutant clones (Fig 1I–K), but not proliferation of wildtype ovarian stem and follicle cells (Fig 1C–C' and E). Suppression of feeding via amino acid

---

**Figure 1. Dietary histidine withdrawal selectively reduces *nerfin-1* and N^ACT but not *pros* clonal growth.**

A    Schematic depicting wildtype NBs (left panel) which express Mira (red), undergoing asymmetric division to generate a GMC, which divides only once to give rise to two postmitotic neurons. The cell fate determinant Pros (blue) is inherited by the GMC, where it translocates to the nucleus to promote differentiation. Postmitotic neurons express Pros (blue) and Nerfin-1 (green). Upon the loss of Nerfin-1 (*nerfin-1⁻*, middle panel), neurons undergo stepwise reversion, by increasing cellular growth, and switching on stem cell genes while maintaining the expression of neuronal-specific markers such as Pros, before their complete reversion to NBs, giving rise to clones consisting a mixture of NBs and neurons. In *pros⁻* clones, GMCs fail to differentiate and revert to NBs, giving rise to clones consisting mostly of Mira⁺ NBs. Schematic depicting type II wildtype NB lineages (left) and dedifferentiation of INP to NBs upon Notch overactivation, which gives rise to clones consisting of mostly NBs (right).

B–E    Representative pictures showing that the withdrawal of dietary leu but not his significantly reduced stem cells and follicle cell proliferation (pH3, red) in the adult ovary after 10 days, quantified in (E) (*n* = 13, 14, 13), scale bar = 100 μm.

F–H    Representative pictures showing that wildtype larval NB clonal growth was significantly reduced after 6 days of dietary leucine depletion (25% leu) and significantly increased by dietary histidine depletion (25% his), quantified in (F) (*n* = 28, 22, 7), scale bar = 50 μm

I–K    Representative pictures showing that adult *nerfin-1* clonal growth was significantly reduced on −his diet compared to CDD (measured at day 0, and day 9) quantified in (K) (*n* = 27, 110, 136). Scale bar = 75 μm. ***P < 0.001

L–N    Representative pictures showing that larval *nerfin-1* clonal growth was significantly reduced on 25% his diet compared to CDD (after 6 days), quantified in (N) (*n* = 53, 36). Scale bar = 50 μm.

O–Q    Representative pictures showing that adult *pros* clonal growth was not significantly altered on −his diet compared to CDD (after 7 days), quantified in (Q) (*n* = 12, 23, 16). Scale bar = 75 μm.

R–T    Representative pictures showing that larval *pros* clonal growth was not significantly altered by dietary histidine reduction (25% his) compared to CDD (after 6 days), quantified in (T) (*n* = 19, 13). Scale bar = 50 μm.

U–W    Representative pictures showing that the growth of larval type II lineages overexpressing N^ACT significantly reduced dietary histidine reduction (25% his) compared to CDD, quantified in (W) (*n* = 27, 19). Scale bar = 100 μm.

Data information: In all graphs, the key indicates that green bars represent a significant increase (*P* < 0.05), red bars a significant decrease (*P* < 0.05), and grey bars no significant change (*P* > 0.05) in *t*-tests with the relevant paired controls (black bar). In all graphs, error bars represent 1 standard error of the mean (SEM). FC, fold change. See also Figs EV1 and EV4.

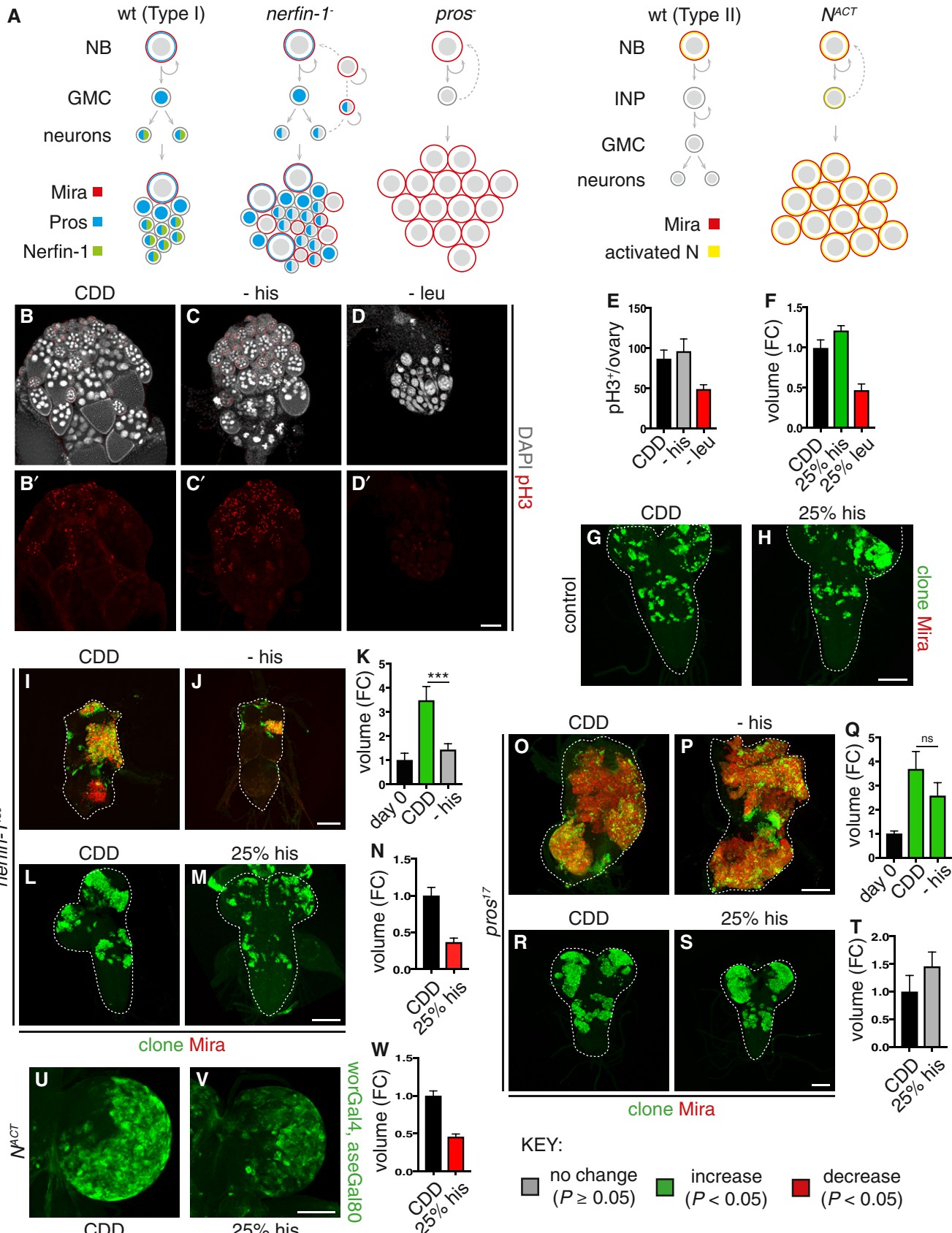

**Figure 1.**

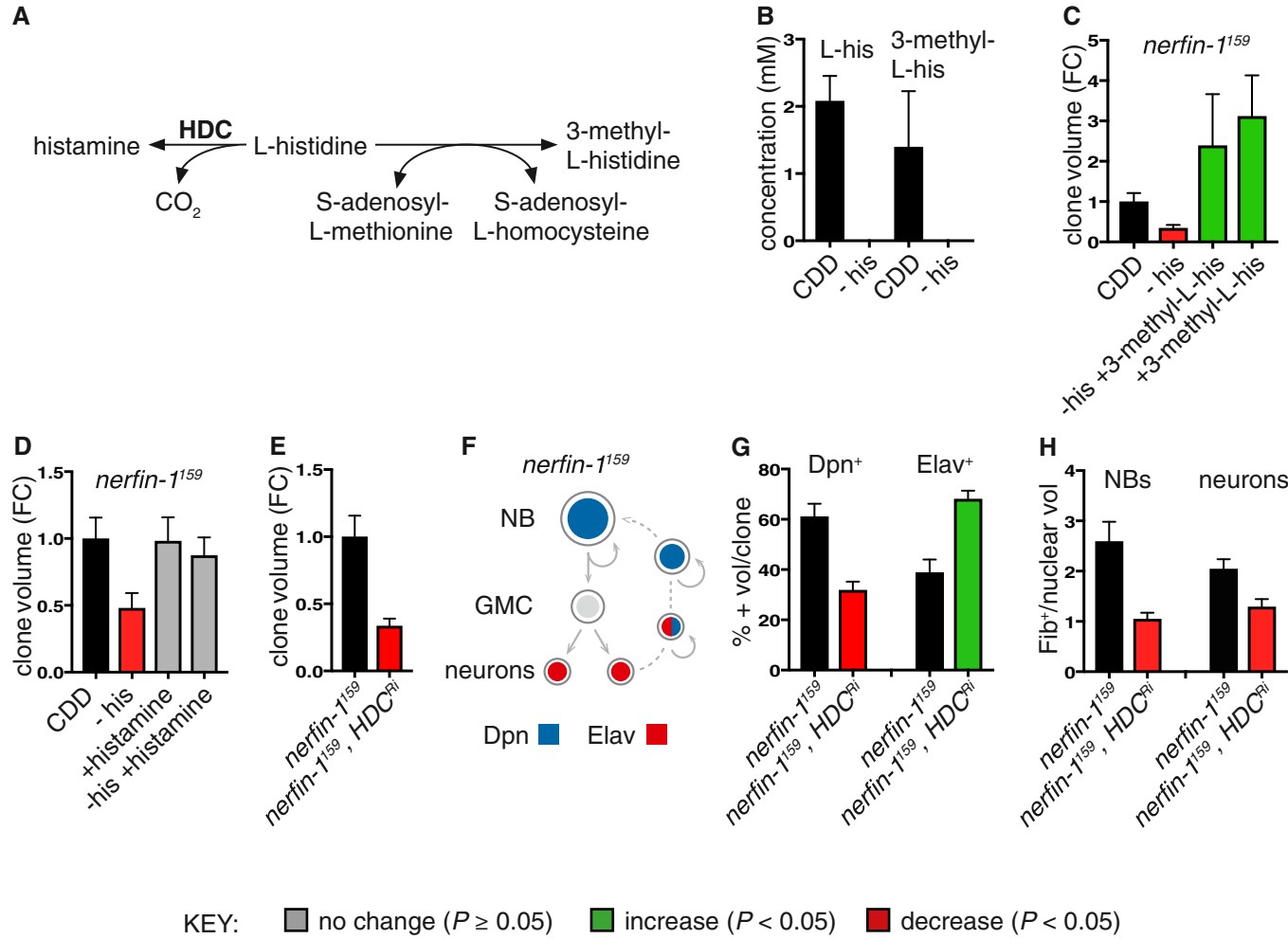

**Figure 2.  Histidine metabolic perturbations affect *nerfin-1* neuron-to-NB reversion.**

A   Schematic depicting the metabolic reaction between histidine and its downstream metabolites histamine and 3-methyl-L-histidine.
B   Concentration of histidine and 3-methyl-L-histidine from whole *Drosophila* adults extracts measured by $^1$H NMR after 3 days of feeding on either CDD or −his diet
     ($n = 3, 3$). Both compounds are reduced to below detection levels ($< 0.3$ mM) upon dietary his depletion.
C   Dietary supplementation of 3-methyl-L-his for 4 days significantly increased adult *nerfin-1* clonal growth and rescued *nerfin-1* growth inhibition due to his dietary
     withdrawal ($n = 18, 14, 7, 9$).
D   Dietary supplementation of histamine does not significantly increase adult *nerfin-1* clone size, but can rescue *nerfin-1* clonal growth inhibition due to his dietary
     withdrawal ($n = 42, 14, 23, 25$).
E   Larval *nerfin-1* clonal growth is significantly decreased upon overexpression of HDC RNAi ($n = 23, 36$).
F   Schematic depicting stepwise neurons to NBs reversion in *nerfin-1* type I lineages where neurons express Elav, NBs express Dpn, and reverting neurons express both
     markers.
G   Hdc inhibition significantly reduced the percentage of Dpn$^+$ NBs ($n = 16, 17$) and significantly increased the percentage of Elav$^+$ neurons per clone ($n = 16, 17$).
H   Hdc$^{Ri}$ overexpression reduced cellular growth of *nerfin-1* NBs ($n = 17, 19$) and neurons ($n = 36, 19$) measured as the ratio of nucleolus/nuclear volume.

Data information: In all graphs, the key indicates that green bars represent a significant increase ($P < 0.05$), red bars a significant decrease ($P < 0.05$), and grey bars no
significant change ($P > 0.05$) in *t*-tests with the relevant paired controls (black bar). In all graphs, error bars represent 1 standard error of the mean (SEM). FC, fold
change. See also Figs EV2 and EV3.

imbalance (Hao *et al*, 2005; Bjordal *et al*, 2014) is unlikely to account for the decreased growth observed, as dye-labelled CDD − his diets were ingested and found in the gut of adult flies (Fig EV1D).

We also developed a second EAA deprivation regime, decreasing the concentration of individual EAAs in the larval diet by 75% (Fig EV1B). Most of the EAAs (e.g. leucine and data not shown) were required for wildtype neural stem cell proliferation (Fig 1F); in contrast, histidine was selectively required by *nerfin-1* mutant clones (Fig 1L–N) but not wildtype neural stem cell proliferation (Fig 1F–H). Together, these results demonstrate *nerfin-1* mutant lineages are more sensitive to dietary histidine depletion than wild-type ovarian and neural stem cells.

To determine whether histidine deprivation affects the growth of some mutant clones more than others, we next tested clones induced via loss of Prospero (Pros) in type I lineages and by Notch activation in type II lineages (Fig 1A). The homeobox transcription factor Pros

has been shown to act as a transcriptional switch for GMC differentiation. In the absence of *pros*, GMCs revert to a stem cell-like state, express neuroblast genes and exhibit hallmarks of cancer such as unregulated proliferative potential and inappropriate differentiation (Bello *et al*, 2006; Betschinger *et al*, 2006; Caussinus & Gonzalez, 2005; Choksi *et al*, 2006; Lee *et al*, 2006; Maurange *et al*, 2008 and Fig 1A). Constitutive Notch (N) activation is responsible for causing overgrowth in type II lineages, resulting in INPs reverting their cell fate back to a stem cell-like state, leading to ectopic expansion of NBs (Bowman *et al*, 2008; Song & Lu, 2011; Weng *et al*, 2010 and Fig 1A). Surprisingly, we observed *pros* clonal growth was insensitive to the complete withdrawal of histidine from the adult diet (Fig 1O–Q) or to a 75% decrease in histidine in the larval diet (Fig 1R–T). In contrast, a 75% decrease in histidine in the larval diet significantly reduced the growth of $N^{ACT}$ clones (Bowman *et al*, 2008; Weng *et al*, 2010) generated with *worGAL4, aseGAL80* (Fig 1U–W).

Thus far, the results demonstrate that the dietary requirement for histidine is much greater for *nerfin-1* and $N^{ACT}$ than for *pros* clonal growth. The differential response to histidine withdrawal does not appear to be directly related to the rate at which mutant clones grow, as $N^{ACT}$ and *pros* clones grow to a similar size (Fig EV1E), yet they exhibit very different sensitivities to histidine manipulations.

### Histidine supplementation and histidine decarboxylase promote *nerfin-1* clonal growth

Nuclear magnetic resonance (NMR) measurements indicated that the histidine-deficient CDD is effective at depleting the majority of free histidine and also a related metabolite, 3-methyl-L-histidine, from whole adult flies (limit of detection for histidine is ~0.3 mM, Figs 2A and B, and EV2A). Histamine, another histidine metabolite known to function as a neurotransmitter (Burg *et al*, 1993), was not detectable by NMR in whole adult flies (therefore estimated to be < 0.3 mM, data not shown) either in the presence or absence of dietary histidine. Dietary histidine is not only rate limiting for *nerfin*-1 (and not *pros*) clonal growth (Fig 1I–T), supplementation of the CDD diet with histidine and its downstream metabolites 3-methyl-L-histidine (but not histamine) is also sufficient to boost *nerfin*-1 (and not *pros*) clonal growth in the adult (Figs 2C and D, and EV2B and C). In addition, in the absence of histidine, dietary supplementation with 3-methyl-L-histidine and histamine can

rescue *nerfin-1* clone size in the adult (Fig 2C and D). Together, this suggests that the effect of histidine on the growth of *nerfin-1* clones is mediated by downstream metabolites 3-methyl-L-histidine and histamine. Interestingly, *nerfin-1* clone volume was not significantly affected by dietary addition of cimetidine (a histamine receptor inhibitor, Hong *et al*, 2006) in the adult or knockdown of HisCl1, a histamine receptor in the larvae (RNAi knockdown efficiency previously quantified in Oh *et al*, 2013; Fig EV3A and B). Together, these results suggest that histamine signalling via its receptor is not required for *nerfin-1* clonal growth. However, when we knocked down CG3454, encoding a predicted histidine decarboxylase (HDC, RNAi knockdown efficiency previously quantified in Oh *et al*, 2013) catalysing the conversion of L-histidine into histamine (Hong *et al*, 2006), we observed a 50% decrease in *nerfin-1* but not wildtype clonal volume (Figs 2E and EV3C), which can be partially rescued by supplementation with dietary histamine (Fig EV3D). Hdc knockdown in *nerfin-1* clones decreased clonal volume concomitant with an inhibition of neuron-to-neuroblast reversion, reflected by a drop in the proportion of Deadpan (Dpn$^+$) NBs and an increase in the proportion of embryonic lethal abnormal vision (Elav$^+$) neurons per clone (Fig 2F and G), without significantly impacting on cell death or cell mitoses (Fig EV3E and F).

### Histidine deprivation inhibits ribosome biogenesis and *nerfin-1* dedifferentiation

Similar to Hdc inhibition, we found histidine-free adult diet significantly decreased the proportion of cells in *nerfin-1* clones that were Mira$^+$ NBs, with a concomitant significant increase in the proportion of Elav$^+$ neurons but no significant change in Dcp1$^+$ apoptotic cells (Fig 3A–E). Similarly, a 75% decrease in dietary histidine during larval development also led to a significant increase in the proportion of Elav$^+$ neurons with a corresponding decrease in the proportion of Mira$^+$ NBs (Fig 3F and G). Together, the larval and adult analyses of *nerfin-1* clones strongly suggest that histidine deprivation skews the balance between stem cell self-renewal and neuronal differentiation in favour of differentiation. Unlike wildtype neurons, a rapid increase in cellular growth reflected by increased ribosome biogenesis is a prerequisite for *nerfin-1* neuron-to-neuroblast dedifferentiation (Froldi *et al*, 2015). As adult dietary histidine withdrawal significantly decreased the diameter of NBs (Fig 3H–J'), we

**Figure 3. Histidine depletion slows down *nerfin-1* dedifferentiation via decreasing cellular growth.**

A–E    Representative images of *nerfin-1* clones in the adult VNC after 6 days of feeding on CDD or −his diet, scale bar = 100 μm. Histidine dietary withdrawal resulted in clones containing significantly more differentiated Elav$^+$ neurons (red) quantified in (D) (*n* = 16, 17) and significantly fewer Mira$^+$ NBs (blue), quantified in (E) (*n* = 12, 22), without significantly altering the amount of cell death per clone quantified in (C) (*n* = 6, 7).

F, G    Larval *nerfin-1* clones consisted of significantly greater proportion of differentiated neurons (*n* = 10, 7) and significantly reduced proportion of NBs per clone upon his dietary depletion (25% his, *n* = 9, 11).

H–P    Representative images of adult *nerfin-1* (I–J', green) and *pros* clones (K–L', green), scale bar = 5 μm. In *nerfin-1* clones, the size of the NBs (Ase$^+$, blue) was significantly reduced upon 6 days of histidine dietary withdrawal, quantified in (H) (*n* = 33, 20). Cellular growth measured as the ratio of nucleolus (Fib, white)/ nuclear (Ase, blue) volume was also significantly reduced, quantified in (M) (*n* = 33, 20). In *pros* clones, the size of the NBs (Ase$^+$ blue) and cellular growth measured as the ratio of nucleolus (Fib, white)/nuclear (Ase, blue) volume were not significantly altered upon 6 days of histidine dietary withdrawal, quantified in (H) (*n* = 9, 8) and (O) (*n* = 9, 7). Cellular growth measured as the ratio of nucleolus/nuclear volume was significantly reduced in larval *nerfin-1* NBs after 6 days his dietary reduction (25% his compared to CDD, *n* = 14, 6) quantified in (N). Panel (P) shows that the larval *nerfin-1* neuron-to-NB reversion requires 15-fold increase in cellular volume (*n* = 10) and *pros* GMC-to-NB reversion requires approximately 3.5-fold increase in cellular volume (*n* = 31).

Data information: In all graphs, the key indicates that green bars represent a significant increase (*P* < 0.05), red bars a significant decrease (*P* < 0.05), and grey bars no significant change (*P* > 0.05) in *t*-tests with the relevant paired controls (black bar). In all graphs, error bars represent 1 standard error of the mean (SEM). FC, fold change. See also Fig EV2.

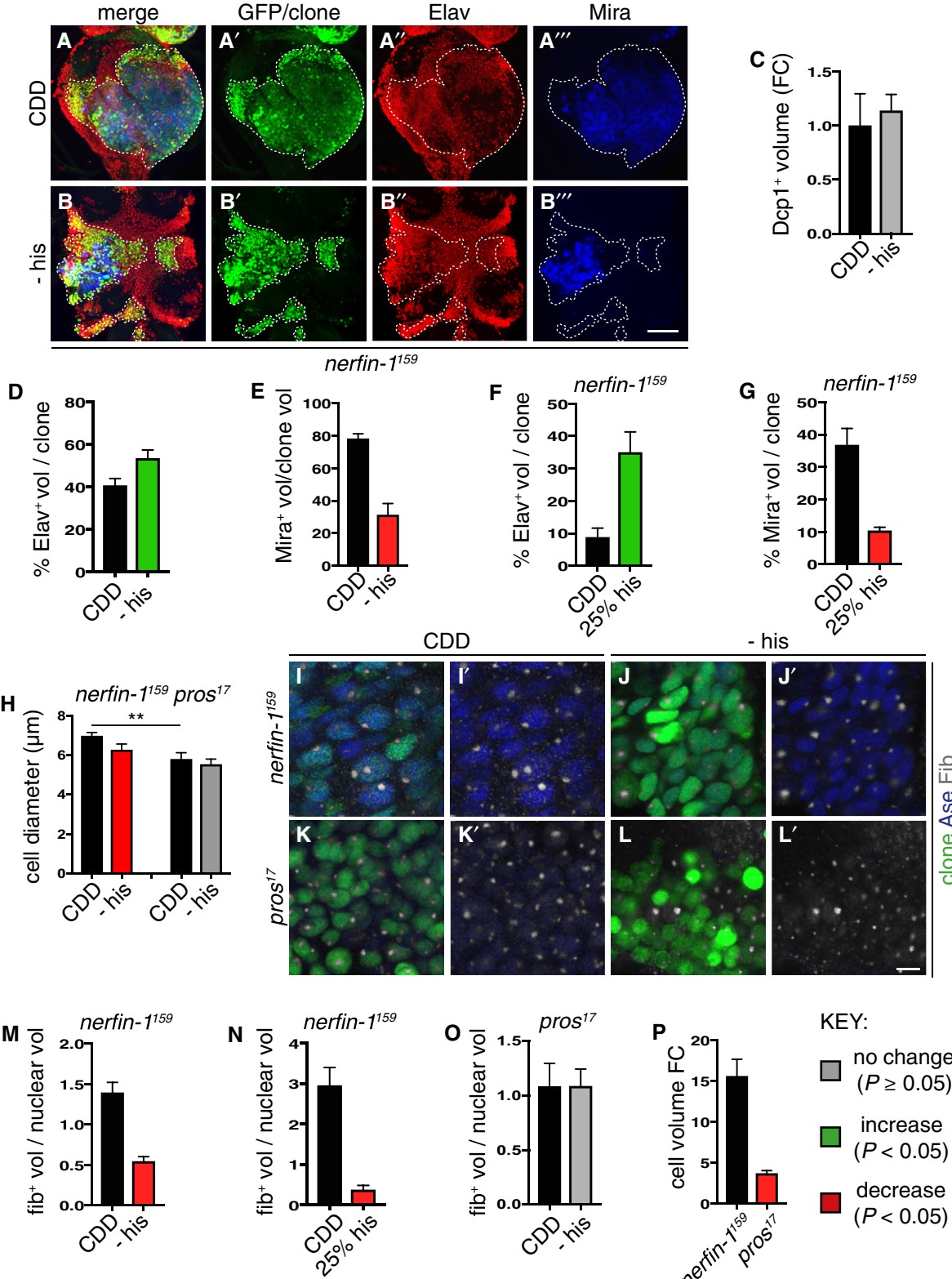

**Figure 3.**

therefore measured nucleolar/nuclear volume ratio as a readout of cellular growth rate (Song & Lu, 2011), using fibrillarin and Asense staining. We observed that, when histidine was withdrawn

completely from the adult diet or was decreased by 75% in the larval diet, NBs in *nerfin-1* clones showed a strong decrease in the nucleolar/nuclear volume ratio (Fig 3I, J, M and N). This was also

recapitulated in the Hdc knockdown setting, where we also observed a significant reduction in nucleolus/nuclear volume ratio in both NBs and neurons (Fig 2H). Therefore, Hdc knockdown or histidine withdrawal both appear to inhibit *nerfin-1* clone growth by altering the self-renewal-differentiation balance via slowing down the rate of neuron-to-neuroblast reversion. In *pros* clones, however, the NB diameter and nucleolar/nuclear ratio were not significantly affected by histidine dietary withdrawal (Fig 3H, K and L). We also found that *nerfin-1* neuron-to-NB reversion involved an increase in cell volume of ~15-fold whereas *pros* GMC-to-NB reversion was associated with only a modest ~3.5-fold increase in cell size (Fig 3P). This is primarily because *pros* NB-like cells tend to be much smaller than their wildtype counterparts (Fig 4A–C). Furthermore, unlike *nerfin-1* NBs which exhibit ~3-fold increase in nucleolus/nuclear volume ratio compared to control (Froldi *et al*, 2015), *pros* NBs are not significantly different from control NBs in terms of cellular growth (Fig 4D). Together, the above results suggest that a major effect of histidine deprivation is to decrease ribosome biogenesis in the nucleolus, which would be expected to decrease cell growth and thus cell size. Hence, the selective effects of histidine deprivation on *nerfin-1* versus *pros* clones may be explained because an increase in cellular growth and cell size is a prerequisite for dedifferentiation in *nerfin-1* clones (Froldi *et al*, 2015) but, for *pros* clones, cellular growth and cell size increases are less relevant.

### Histidine deprivation inhibits *nerfin-1* dedifferentiation via Myc and eIF4E

Increased cellular growth and size are mediated by two key regulators: Tor (Target of Rapamycin) and Myc. Tor signalling allows animals to regulate growth and energy balance in response to fluctuations in dietary amino acids and other environmental conditions (Hennig *et al*, 2006). Myc is a key Tor effector and controls cell growth, at least in part by regulating ribosome biogenesis via the transcriptional control of rRNA, ribosomal proteins and other proteins (Teleman *et al*, 2008; van Riggelen *et al*, 2010). To compare directly the requirement of Myc by *nerfin-1* and *pros* clones, we generated *pros* and *nerfin-1* clones in a *myc[PO]* heterozygous background. This manipulation significantly reduced *nerfin-1* clone size,

nucleolar/nuclear ratio and the proportion of Dpn[+] NBs in *nerfin-1* clones, with a concomitant increase in the proportion of Elav[+] neurons (Fig 4J–N). In contrast, *pros* clone size, nucleolar/nuclear ratio and the proportion of NBs and neurons were not significantly altered (Fig 4E–I). This finding indicates that *myc* gene dosage is more critical for *nerfin-1* than for *pros* clonal growth. Such a differential Myc requirement is consistent with the preceding nucleolar and cell size measurements and may reflect the larger amount of cell growth during the neuron-to-NB dedifferentiation in *nerfin-1* clones, compared to the GMC-to-NB dedifferentiation in *pros* clones.

We next investigated whether altering Myc levels could affect how *nerfin-1* clones respond to changes in dietary histidine. Strikingly, *nerfin-1* clones generated in a *myc* hypomorphic background (*myc[PO]* heterozygotes) fail to respond to dietary histidine supplementation (Fig 4O). Moreover, overexpression of Myc was able to rescue the growth of *nerfin-1* clones due to a 75% decrease in dietary histidine (Fig 4P). Upon Hdc knockdown, we observed both a significant decrease in the percentage of Myc-expressing cells and Myc-expressing Elav[+] neurons (an early marker of dedifferentiation) per clone (Fig EV3G–I). Together, these findings show that the sensitivity of *nerfin-1* clones to histidine perturbations is dictated by clonal Myc levels. More specifically, dietary histidine withdrawal impacts on *nerfin-1* clonal growth by downregulating Myc, whereas overactivation or suppression of Myc can both render *nerfin-1* clones insensitive to histidine.

The Tor pathway is a key mediator of the response to dietary amino acids and, like Myc, promotes cellular growth and ribosome biogenesis (Hietakangas & Cohen, 2009). We found that overexpressing a dominant negative form of Tor (*Tor[TED]*) significantly affected both *nerfin-1* and *pros* clonal growth (Fig 4Q and R). Interestingly, while *Tor[TED]* decreases *nerfin-1* clone volume via reducing the ratio of nucleolus/nuclear volume and cellular growth (Froldi *et al*, 2015), *Tor[TED]* did not significantly alter the rate of cellular growth in *pros* NBs (Fig 4D). Furthermore, an antisense transgene against a putative amino acid transporter Slimfast (Slif), which is known to act upstream of Tor (Colombani *et al*, 2003), also differentially affected *nerfin-1* and *pros* mutant clones (Fig 4Q and R). Tor controls growth in part by phosphorylation of ribosomal protein S6 kinase (S6K) and eIF4E binding protein (4E-BP) which, in turn, promote translation

**Figure 4.  Histidine deprivation inhibits *nerfin-1* clones via Myc and eIF4E.**

A–D   Representative pictures showing that *pros* NBs and GMCs are smaller than wildtype (NBs and GMCs are distinguished at telophase, as a doublet of unequal size), quantified in (C) (*n* = 25, 31, 21, 16). *pros* NBs exhibit similar nucleolus/nuclear ratio (white arrows, nucleolus marked by Fib) compared to wildtype and *pros;Tor[DN]* NBs, quantified in (D) (*n* = 10, 18, 15).

E–I   Representative pictures showing *pros* clones (green) generated in *myc* hypomorphic background did not significantly alter *pros* nucleolus/nuclear ratio (E, F), nucleolus labelled with Fib (red) [quantified in (G) (*n* = 44, 43)], clone size [quantified in (H) (*n* = 41, 34)] or the proportion of Dpn[+] NBs and Elav[+] neurons per clone [quantified in (I) (*n* = 15, 15, 15, 15)].

J–N   Representative pictures showing that *nerfin-1* clones (green) generated in *myc* hypomorphic background (J, K) exhibited significantly reduced nucleolus/nuclear ratio, [nucleolus labelled with Fib (red)] and reduced clone size, quantified in (L) (*n* = 34, 27) and (M) (*n* = 106, 75), and displayed an increased proportion of differentiated Elav[+] neurons and reduced proportion of Dpn[+] NBs per clone, quantified in (N) (*n* = 20, 20, 20, 20). scale bar = 10 μm

O   Reducing Myc dosage with the hypomorphic Myc allele, *myc[PO]*, abolished the increase in *nerfin-1* clone size mediated by 2× histidine dietary intake, 2× his (*n* = 41, 18, 10, 57).

P   Overexpression of *myc* restored the growth of *nerfin-1* clones inhibited by histidine dietary reduction, 25% his (*n* = 51, 36, 39, 29).

Q, R   *pros* (Q) and *nerfin-1* (R) clones are reduced upon Tor inhibition (*n* = 22, 63) compared to control (*n* = 51, 65), but their growth differentially depends on amino acid transporter *Slif* (*n* = 12, 32) and downstream growth regulators eIF4E (*n* = 21, 47) and S6K (*n* = 39, 37).

Data information: In all graphs, the key indicates that green bars represent a significant increase (*P* < 0.05), red bars a significant decrease (*P* < 0.05), and grey bars no significant change (*P* > 0.05) in *t*-tests with the relevant paired controls (black bar). In all graphs, error bars represent 1 standard error of the mean (SEM). FC, fold change. See also Fig EV3.

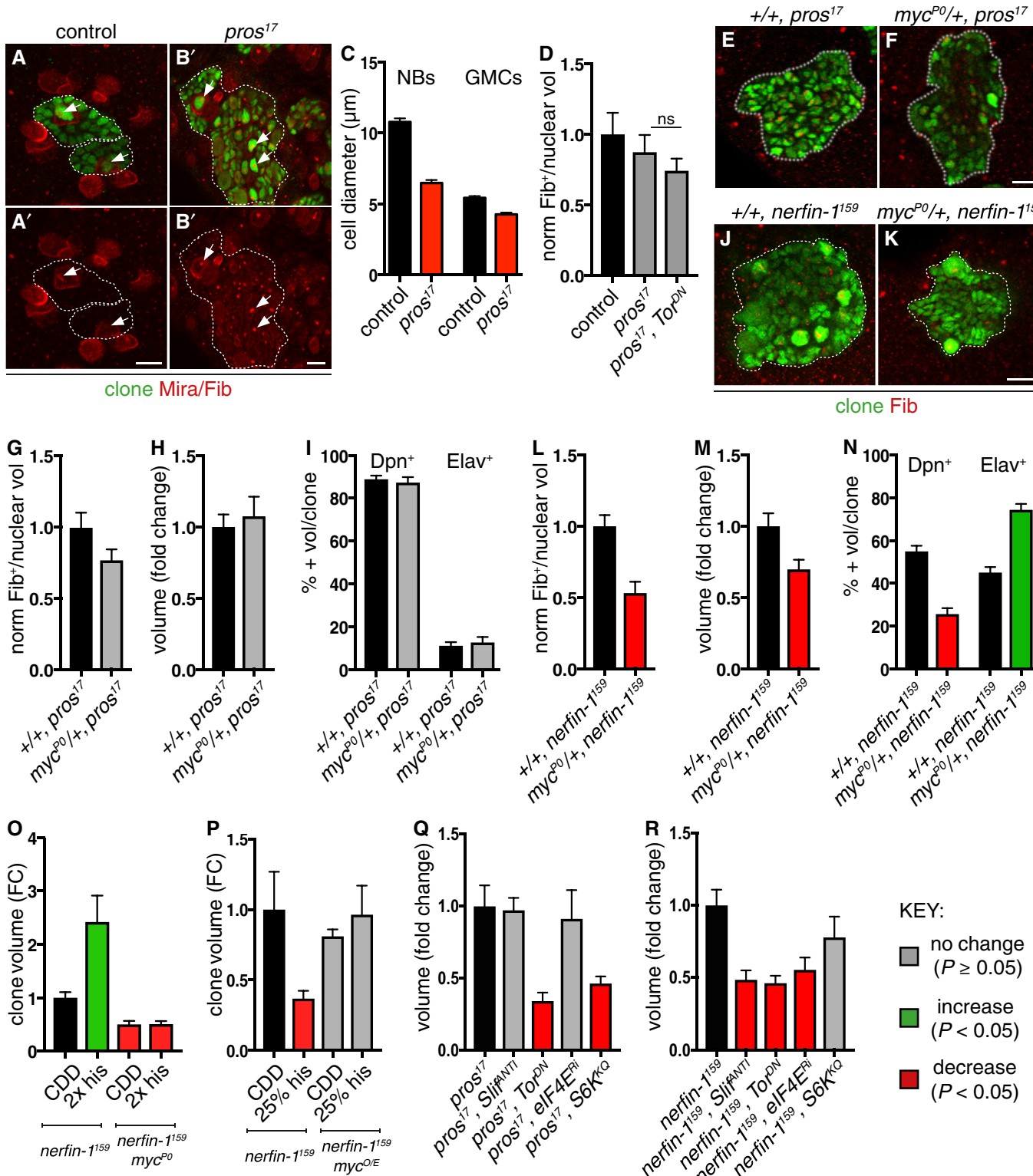

**Figure 4.**

and ribosome biogenesis by regulating ribosomal protein S6 (RpS6) and eIF4E, respectively (Miron *et al*, 2003). We found that overexpression of a dominant negative form of S6K significantly decreased *pros* but not *nerfin-1* clonal growth (Fig 4Q and R). Conversely,

*nerfin-1* but not *pros* clone size was reduced by eIF4E RNAi knockdown (Fig 4Q and R). This shows that *nerfin-1* and *pros* mutant clones have a differential growth requirement for S6K and eIF4E, and Tor targets involved in translation and ribosome biogenesis. Like

*nerfin-1* clones, the growth of neural $N^{ACT}$ clones is also dependent on the growth regulators Myc and eIF4E (Song & Lu, 2011). Hdc knockdown in $N^{ACT}$ clones significantly decreased the growth of type II lineages (identified by their absence of Ase, Fig EV4A–C). These results demonstrate a strong growth requirement for Myc and eIF4E in two different mutant neural lineages (*nerfin-1* and $N^{ACT}$) that are highly sensitive to dietary histidine, whereas this is not the case for relatively histidine-insensitive *pros* clones.

To investigate tissue-of-origin effects on histidine sensitivity, we compared $N^{ACT}$ clones in the CNS with those in the epithelium of the eye imaginal disc (Baonza & Garcia-Bellido, 2000; Brumby, 2003; Estella & Baonza, 2015). We observed that the growth of $N^{ACT}$ clones in the eye discs which exhibited dependency on Myc was also significantly decreased by Hdc knockdown (Fig EV4D–G). Therefore, despite the fact that N targets in neuroblast lineages differ from those in the epithelium (Djiane *et al*, 2013; Zacharioudaki *et al*, 2016), and their division mode differs, both mutant clone types are sensitive to histidine manipulations. In contrast, we observed that the growth of eye epithelial clones induced by loss-of-function of the tumour suppressor *scribble* (*scrib*[1]) in combination with activated Ras (*Ras*$^{ACT}$; Brumby, 2003), the growth of which is not dependent on Myc (data not shown), was unaffected by Hdc inhibition (Fig EV4H–J). We therefore conclude that in both neuroblast and eye discs, the growth of mutant clones dependent on Myc also exhibits sensitivity to histidine perturbations.

This study reveals that *nerfin-1*, $N^{ACT}$ and *pros* clones in the CNS have different sensitivities to perturbations in dietary histidine and its metabolism. The demand for histidine is not simply correlated with the overall level of growth because clones of similar sizes, such as *pros* and $N^{ACT}$, display different degrees of sensitivity to histidine perturbations. A major finding is that the histidine sensitivity of these mutant neural lineages is dictated by their dependence upon the cell growth and ribosome biogenesis components Myc and eIF4E. Hence, we found that increased nucleolar size and high Myc dependence are good predictors of whether mutant neural lineages will respond to dietary and genetic histidine perturbations. The strong requirement for dietary histidine in *nerfin-1* but not *pros* mutant clones thus reflects the high dependence upon the Myc-mediated cell size increase that is essential for the neuron-to-NB dedifferentiation process that drives overall growth and proliferation (Froldi *et al*, 2015). Like *nerfin-1* clones, we found that the growth of $N^{ACT}$ clones is sensitive to histidine. It is interesting that this also correlates with a dependence upon Myc and eIF4E and occurs via a related dedifferentiation mode of growth: in this case, from INPs to NBs (Song & Lu, 2011). In contrast, we showed that *pros* mutant clones are relatively insensitive to histidine manipulations and this correlates with a low sensitivity to Myc and a relatively modest size increase during GMC-to-NB dedifferentiation. Although both *nerfin-1* and *pros* clones require activation of Tor kinase, a master regulator that couples amino acid availability to cell growth, downstream of Tor, *nerfin-1* but not *pros* clones, requires the CAP-dependent translation regulator eIF4E. Therefore, histidine metabolic perturbations appear to affect the growth of *nerfin-1* (and $N^{ACT}$) and not *pros* mutant clones due to their differential reliance on cellular growth and ribosome biogenesis. Interestingly, the overall size of the mutant clones does not correlate with cellular growth, such that

*nerfin-1* clones are smaller and slower growing than *pros* clones, yet they have a greater reliance on cellular growth.

In conclusion, histidine perturbations target the nucleolar and cellular growth of clones that are driven by a Myc-dependent mode of growth involving dedifferentiation. Future work, beyond the scope of this study, will be needed to decipher the molecular details of how histidine metabolism regulates Myc-dependent cell growth. Whatever the pathways involved, this study establishes the principle that histidine metabolism represents a metabolic vulnerability for some Myc-dependent cancers that might be therapeutically harnessed by dietary interventions or by targeting histidine transporters or enzymes.

# Materials and Methods

### Dietary manipulations

Adult dietary manipulation to evaluate clone or ovary growth was carried out by rearing larvae on standard *Drosophila* medium (Cheng *et al*, 2011) and transferring pupae onto CDD, CDD-EAAs, CDD + his, CDD + histamine or CDD + 3-methyl-L-histidine diets, and clones or ovaries were evaluated at various days after adult hatching. All experiments were carried out at 25°. Nutrient restriction in the adult was carried out by feeding newly hatched flies on standard *Drosophila* medium or 5% agarose in PBS (Sigma), and clone size was evaluated at 3 days after feeding or nutrient restriction. Larval dietary manipulation was carried out by feeding larvae on CDD or CDD with reduced concentrations of histidine or leucine for 8 days. For assessing the effect of metabolic intermediates on clone growth, 500 mg/ml cimetidine (Sigma-Aldrich c4522), 250 mM histamine (Sigma-Aldrich H7125) and 12 mM 3-methyl-L-histidine (Sigma-Aldrich M9005) were added to CDD.

### Chemically defined diet (CDD)

Chemically defined diet has the following composition: L-Alanine (5.612 mM, Sigma-Aldrich A7627), L-Arginine (4.592 mM, Sigma-Aldrich A5006), L-Aspartic Acid (3.756 mM, Sigma-Aldrich A9249), L-Cysteine (4.127 mM, Sigma-Aldrich C7352), L-Glutamic Acid (36.702 mM, Sigma-Aldrich G1251), Glycine (6.661 mM, Sigma-Aldrich V900144), L-Histidine (6.445 mM, Sigma-Aldrich H8125), L-Isoleucine (22.869 mM, Sigma-Aldrich I2752), L-Leucine (15.247 mM, Sigma-Aldrich L8000), L-Lysine HCl (12.997 mM, Sigma-Aldrich 23128), L-Methionine (5.362 mM, Sigma-Aldrich M9625), L-Phenylalanine (7.8 mM, Sigma-Aldrich P2126), L-Proline (4.343 mM, Sigma-Aldrich P0380), L-serine (4.758 mM, Sigma-Aldrich S4500), L-Threonine (16.790 mM, Sigma-Aldrich T8625), L-Tryptophan (2.448 mM, Sigma-Aldrich T0254), L-Tyrosine (2.760 mM, Sigma-Aldrich I2752) and L-Valine (23.901 mM, Sigma-Aldrich V0500). Cholesterol (0.2 g/l, Sigma-Aldrich C8667) dissolved in Nipagin/Bavistin solution (19.64 g/l), Adenosine-5′(3′)-monophosphate (Santa Cruz, SC-479689, 0.6 g/l), Guanosine-5′(3′)-monophosphate (Santa Cruz, SC-295032, 0.4 g/l), Uridine-5′(3′)-monophosphate (Santa Cruz, SC-22403, 0.4 g/l), Cytidine-5′(3′)-monophosphate (Santa Cruz, SC-211159, 0.4 g/l), NaHCO$_3$ (Sigma-Aldrich S5761, 1 g/l), KH$_2$PO$_4$ (Sigma-Aldrich P5655, 0.71 g/l), K$_2$HPO$_4$ (Sigma-Aldrich P3786, 3.733 g/l), MgSO$_4$.7H$_2$O (Sigma-

Aldrich M1880, 0.82 g/l), NaCl (Sigma-Aldrich S7653, 0.04 g/l), Fe.Na EDTA (Sigma, EDFS, 0.02 g/l), Zn.Na EDTA (Fluka, 34553, 0.02 g/l), Mn.NA EDTA (Alfa Aesar, 40519, 0.02 g/l), Cu.Na EDTA (Fluka, 03668, 0.005 g/l), Agarose (Sigma-Aldrich A9054, 7 g/l) and Sucrose (Sigma-Aldrich S9378, 10 g/l) were boiled and dissolved, and then cooled down to 42°C before the heat-sensitive solution was added (see below), and poured into individual vials. The heat-sensitive solution consists of Choline Chloride (Sigma-Aldrich C7527, 0.06 g/l), Thymidine (Sigma-Aldrich T9656, 0.2 g/l), Calcium Gluconate (Sigma-Aldrich C8231, 0.05 g/l), Thiamine Hydrochloride (Sigma-Aldrich T1270, 0.002 g/l), Riboflavin (Sigma-Aldrich R9504, 0.01g/l), Nicotinic Acid (Sigma-Aldrich N0761, 0.012 g/l), Calcium Pantothenate (Sigma-Aldrich P5155, 0.016 g/l), Pyridoxine Hydroxylchloride (Sigma-Aldrich P6280, 0.0025 g/l), Folic Acid (Sigma-Aldrich F8758, 0.003 g/l), DL-Carnitine Hydrochloride (Sigma-Aldrich 729868, 0.001 g/l), Biotin (Sigma-Aldrich B4639, 0.0002 g/l), Penicillin (Sigma-Aldrich P3032, 0.25 g/l) and Streptomycin (Sigma-Aldrich S2522, 0.25 g/l) dissolved in $H_2O$.

## Fly strains

We used the following strains for generating CNS and eye imaginal disc MARCM clones (Lee & Luo, 2001): (i) (3L) *w, tub-Gal4, UAS-nlsGFP::6xmyc::NLS, hs-flp; FRT2A, tubP-Gal80LL9/TM6b*; (ii) (3R) *w, tub-Gal4, UAS-nlsGFP::6xmyc::NLS, hs-flp; FRT82B, tubP-Gal80 LL3/TM6b*; (iii) *w;; FRT2A* was used to generate control MARCM clones; (iv) *w;; FRT82B, pros$^{17}$/TM6B* to generate *pros$^{17}$* clones (Bloomington); (v) *w ey-FLP1,UAS-mCD8-GFP;;tub-GAL4 FRT82B tub-GAL80*; (vi) *w;; FRT2A, Df(3L)nerfin-1$^{159}$/TM6b* (Kuzin *et al*, 2005) was used to generate *nerfin-1$^{159}$* clones; (vii) *w;; FRT82B, UAS-N$^{ACT}$* (Kidd *et al*, 1998; Song & Lu, 2012) was used to generate N$^{ACT}$ clones; and (viii) *w;;FRT82B scrib$^1$;UAS-Ras$^{V12}$* (Brumby, 2003) was used to generate *scrib$^1$;Ras$^{V12}$* clones. Other genetic elements used are as follows: *w, UAS-Dcr2; wor-Gal4, ase-Gal80/ CyO; UAS-CD8::GFP* (Bowman *et al*, 2008); *UAS-CG3454 RNAi* (HDC$^{Ri}$, VDRC 34621); *UAS-myc RNAi* (VDRC 106066); *UAS-myc* (BL9674); *UAS-tor$^{TED}$* (Hennig & Neufeld, 2002); *UAS-lacZ RNAi* (VDRC 51446); *UAS-HisCl1 RNAi* (VDRC 104966); *UAS-S6K$^{KQ}$* (BL6911); *UAS-eIF4E RNAi* (VDRC 100722); *myc$^{P0}$* (Quinn *et al*, 2004); and *UAS-Slif$^{ANTI}$* (Colombani *et al*, 2003). The number of transgenes is kept constant in all experiments.

## Immunostaining

Larval and adult tissues were fixed for 20 min in 4% formaldehyde in PBS and immunostained as previously described (Bello *et al*, 2003). The primary antibodies used were anti-Mira (mouse, 1:50, gift of F. Matzusaki), anti-GFP (chick, 1:2,000, Abcam), anti-pH3 (rat, 1:500, Abcam), anti-Dpn (rabbit, 1:100, gift of Y.N. Jan), anti-Dpn (guinea pig, 1:1,000, gift of James Skeath), anti-Ase (rabbit, 1:50, gift of F. Matsuzaki), anti-Elav (mouse or rat, 1:100, Developmental Studies Hybridoma Bank), anti-Myc (rabbit, 1:100, Santa Cruz) and anti-Fib (mouse, 1:200, Abcam). Secondary goat antibodies conjugated to Alexa488, Alexa568, Alexa650 and Alexa505 (Molecular Probes) were used 1:200. DAPI (Molecular Probes) was used at 1:10,000. Fluorescent images were collected on a Leica SP5 confocal microscope, and all images shown are single sections unless otherwise stated.

## Clone induction

For larval CNS dissections, larvae were heat-shocked at 24 h ALH at 37°C for 8 min (*pros*) or 12 min (*nerfin-1* and N$^{ACT}$) and dissected 96 h later. For larval eye imaginal disc dissections, larvae were heat-shocked at 24 h ALH at 37°C for 15 min (N$^{ACT}$ and *scrib$^1$; Ras$^{V12}$*). For adult dissections, larvae were heat-shocked at 72 h ALH at 37°C for 8 min (*pros*), 10 min or 12 min (*nerfin-1*) except in Fig EV1C, where larvae were heat-shocked at 48 h ALH for 1 h and dissected at 9 days after eclosion. Heat shock did not affect the length of larval development, and experimental and control groups in each individual experiment were always heat-shocked and dissected at the same time point. Type II clones were identified by the presence of Dpn$^+$Ase$^-$ NBs.

## Volume measurements

Clone volume of NB lineages was measured from 3D reconstructions of 1.5 μm spaced confocal Z stacks with Volocity software (PerkinElmer) or Imaris (Bitplane). Cellular volume, nuclear and nucleolar volumes were estimated with the formula $4/3\pi r^3$, with r measured from single confocal sections using the Leica LAS software to average orthogonal measurements of cell diameter (2r). NB diameter was measured from single confocal section using Volocity software as the average of orthogonal measurements. Eye disc clone volume was measured from 3D constructions of 1.5 μm spaced confocal Z stacks with Volocity software (PerkinElmer) or Imaris (Bitplane).

## Statistical analysis

*P*-values are calculated by performing two-tailed, unpaired Student's *t*-test. The Welch's correction was applied in case of unequal variances. A non-parametric test (Mann–Whitney test) was used when data showed significant deviation from a normal distribution.

In all graphs, the key indicates that green bars represent a significant increase ($P < 0.05$), red bars a significant decrease ($P < 0.05$), and grey bars no significant change ($P > 0.05$) compared to the relevant controls. Error bars represent 1 standard error of the mean (SEM). All values were calculated as fold change of control except otherwise stated.

## Cell cycle speed

Cell cycle speed was calculated as percentage of cells in M phase, by measuring the number of pH3 +ve NBs as a percentage of total NBs per clone. In nerfin-1 clones, cell cycle speed was measured separately for NBs > 8 μm (fully dedifferentiated NBs) and NBs < 8 μm (dedifferentiating neurons), as we have previously shown (Froldi *et al*, 2015) that NBs cycle faster once fully dedifferentiated.

## NMR

For NMR metabolite evaluation, adults were eclosed onto CDD (0.5 g/l his) or CDD −his. After 3 days on these diets, flies were transferred to CDD −his for 4.5 h to deplete their gut content of food with histidine and to replenish it with food free from histidine. This gut clearance was assessed by feeding adults with food

containing the blue dye Bromophenol Blue sodium salt (Sigma-Aldrich), and after 4.5 h, all the coloured food was cleared from the gut. Methanol–chloroform metabolite extraction was performed on homogenised samples of 15 male or female adult flies; using the method of Bligh and Dyer (Bligh & Dyer, 1959), the experiment was repeated three times. NMR spectra were acquired using Bruker Avance III HD instruments with a nominal 1H frequency of 700 or 800 MHz using 3-mm tubes in a 5-mm CPTCI cryoprobe. For 1H 1D profiling spectra, the Bruker pulse program *noesgppr1d* was used with a 1-s presaturation pulse (50 Hz bandwidth) centred on the water resonance, 0.1-ms mixing time and 4-s acquisition time at 25°C. Typically, 512 transients were acquired. The metabolite concentrations were measured using the *Chenomx NMR Suite 8.1* (Chenomx Inc.) according to the software's standard procedures. The data sets were apodised, line broadened to 0.3 Hz, zero-filled, Fourier-transformed, phase-corrected and spline baseline-corrected in *Chenomx NMR Processor*. The chemical shape indicator was fitted to the 4,4-dimethyl-4-silapentane-1-sulphonic acid (DSS) trimethylsilyl resonance, and the spectra were transferred to *Chenomx NMR Profiler*. The library spectra were fitted in a semi-automated fashion to the experimental data, yielding a readout of the metabolite concentration relative to that of the known DSS standard. Volumes of adult whole fly homogenate were determined using the volume determination with two standards technique, and absolute metabolite concentrations were back-calculated from these volumes as described by Ragan *et al* (2013).

**Expanded View** for this article is available online.

## Acknowledgements

We are grateful to Paul Driscoll of the Francis Crick Institute Science Technology Platform for Metabolomics for advice and acknowledge Tom Frenkiel of the MRC Biomedical NMR Centre. We also thank Rita Sousa-Nunes and Andrew Bailey for critical reading of the manuscript. F.F., O.C., M.Z. and L.Y.C. are supported by NHMRC grant APP1044704 and Peter MacCallum Cancer Institute start-up funding. P.P., T.F. and A.P.G. are supported by the Francis Crick Institute, which receives its core funding from Cancer Research UK (FC001088), the UK Medical Research Council (FC001088) and the Wellcome Trust (FC001088), and previously by the UK Medical Research Council, National Institute for Medical Research (U117584237).

## Author contributions

FF, MS and LYC contributed to the conception of the work. FF, PP, MS, OC, TF, APG and LYC contributed to the experimental design, acquisition and analysis of the work. APG and LYC drafted and revised the manuscript.

## Conflict of interest

The authors declare that they have no conflict of interest.

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
