## [Review Process File · The EMBO Journal]

Histidine is selectively required for the growth of Myc-dependent dedifferentiation tumours in the *Drosophila* CNS

Francesca Froidi, Panayotis Pachnis, Milán Szuperák, Olivia Costas, Tharindu Fernando, Alex P Gould, Louise Y Cheng

Review timeline:	Submission date:	23rd May 2018
	Editorial Decision:	26th Jun 2018
	Revision received:	29th Oct 2018
	Editorial Decision:	26th Nov 2018
	Revision received:	7th Dec 2018
	Editorial Decision:	9th Jan 2019
	Revision received:	9th Jan 2019
	Accepted:	16th Jan 2019

Editor: Karin Dumstrei

Transaction Report:

1st Editorial Decision

26th Jun 2018

Thank you for submitting your manuscript for consideration by the EMBO Journal. Your study has now been seen by two referees and their comments are provided below.

Both referees find the analysis interesting, however further work is also needed to consider publication here. In particular referee #2 raises a number of relevant concerns that should be addressed. The concerns raised are clearly outlined below. Should you be able to address the concerns raised in full then we would be interested in considering a revised version. I should add that it is EMBO Journal policy to allow only a single major round of revision, and that it is therefore important to address the raised concerns at this stage. acceptance of your manuscript will therefore depend on the completeness of your responses in this revised version.

REFeree REPORTS:

Referee #1:

Cheng and colleagues have nicely identified, with use of a holidic chemically defined diet and the powerful genetics of *Drosophila*, a specific requirement of the aminoacid Histidine to the growth of neural clones that overgrow as a consequence of de-differentiation of neurons. Authors present evidence that this requirement is specific to mutant clones that rely on the activity of the dMyc proto-oncogene. Overall, this manuscript deals with a highly relevant and timely topic nowadays, tumour-specific dependence on specific nutrients/aminoacids, and such, it will open new avenues towards the understanding of the interplay between cancer and metabolism. This ms is also well written, figures self-explanatory and the mechanistic understanding of Histidine dependence is

beautifully addressed with clean and well-designed experiments. I consider this ms, then, a strong candidate for EMBO Journal.

The following "minor" issues should be addressed in order to improve the quality of the ms and increase the potential numbers of readers:

(1) The word "tumour" should be used in a cautious manner. Are N-act expressing clones tumours, or just overgrowing clones? Are nerfin mutant clones in Figure 1-4 tumours? Or just clones of cells that have de-differentiated and are overgrowing? I would try to go through the ms and reduce the tone in this regard and call a clone what is a clone and a tumour what is a tumour. Perhaps a small intro to explain that these clones would give rise to tumours is needed, but what is presented in all figures are clones.

(2) The first page of the results is a little bit complicated to follow and might need a little bit of re-phrasing for the non-Drosophila and/or non-NB people. The fact that clones are induced in larval development and analysed in adulthood might not very popular among all Drosophilists and might need a couple of sentences to add the corresponding references and explain the logic behind. Second, authors use Leucine as control. Why Leucine? This should be better explain. Third, authors identify a requirement of nerfin-1 mutant clones for Histidine and other EAAs except Valine. Why do then authors focus on His? Again, a small explanation is needed.

(3) In page 7, the wording used to explain the differential impact of addition of histidine relatives to the growth of nerfin clones should be improved. The fact that histamine supplementation is able to rescue the loss of histidine should be better explained. At the end of the paragraph, "showed to be" and not "showed be".

(4) In page 9, the first sentence says exactly the opposite as it is been shown. I imagine this is simple wording mistake.

(5) The use of the slimfast antisense transgene should be validated with the use of an RNAi transgene.

Referee #2:

In this work, the authors investigate the effect of diet on the growth of various tumors in *Drosophila melanogaster*. For this purpose, they first develop a chemically defined diet. By individually removing or reducing 9 essential amino acids from this diet, they then identify histidine as being largely dispensable for the development of ovaries or neuroblast clones in the adult, but being limiting for the pathological growth of nerfin-1 mutant NB1- and NB2-derived tumors. A similar dependency on histidine is shown for Notch-induced tumors of the NB2 lineage, but not for prospero-mutant tumors of the NB1 lineage or epithelial tumors induced by mutant Ras. The authors further provide evidence that the histidine-dependency involves its metabolite histamine, and that it is phenocopied by knockdown of the growth regulators Myc or eIF4E, and overcome by Myc overexpression. Taken together, this study suggests that some tumors (but not all) strongly rely on histidine derivatives for their nucleolar, cellular and overall growth.

The observation that histamine (which to my knowledge cannot be converted to histidine in animal cells) is required for the efficient proliferation of certain tumors is potentially very interesting - it suggests that some function of histidine other than its incorporation in proteins is important for tumor growth. However, the rationale underlying this work is not quite clear to me. In the examples cited in the introduction, certain normally non-essential amino acids (such as glutamine) become essential for some tumors, thereby revealing a (potentially) exploitable vulnerability of these tumors. Here, on the other hand, the authors start out with amino acids that are thought to be essential for all cells and unexpectedly find that histidine is not essential in some situations (they do not mention whether larvae would survive in the complete absence of histidine). To me, this primarily raises the question why normal *Drosophila* neuroblasts and ovaries can cope without histidine. Supposedly animal cells are incapable of synthesizing histidine - so how can these cells keep making proteins to proliferate? If this were a particularity of these *Drosophila* cells, the impact of the present findings would be limited.

Specific criticisms & suggestions:

1. In flies fed a histidine-free diet, the "missing histidine" could be provided by other organs (e.g. by the fat body) or by gut microbes. The latter possibility should be addressed by testing the histidine-

dependence in bacteria-free (i.e. antibiotics treated) flies.

2. It is not clear whether Myc-overexpression acts tumor cell-autonomously to rescue the growth of "nerfin-1 HDC-knockdown" tumors. In the corresponding experiment the authors use hs-FLP to induce clones throughout the animal - not just in the nervous system. It is well conceivable that Myc-overexpression in some other tissue has a non-autonomous impact on tumor cell growth, as Myc has been shown before to have non-autonomous effects on animal growth. The authors should repeat some of these experiments with the NBII tumor system (driven by "wor-GAL4 ase-GAL80") to reduce such possibly confounding effects.

A similar explanation can also be applied to the Myc reduction-of-function experiments, where the whole animal is Myc[P0/+], not just the tumor cells. These latter experiments have the additional caveat that the experimental Myc[P0/+] animals were females, whereas the controls they were compared to [Y/+] were males - and sex has been reported to affect tissue growth in flies, therefore could affect tumor growth here, too.

3. Myc manipulated flies are characterized in terms of nucleolar, nuclear and tumor size. The observed effects might have been expected, since Myc is known to affect these parameters in many non-transformed cells as well. The authors should determine the composition of the resulting tumors (percentage of NBs, GMCs, differentiated neurons) to determine whether Myc also affects the nature of the tumors.

4. It is not clear to which extent the various regimes of larval clone induction affect the duration of development until adult eclosion - it is not stated in the Methods section that this duration was controlled. Alterations could of course affect tumor size and make it hard / impossible to compare the different experiments.

5. The statement that the histamine receptor HisC11 is not involved in the observed "histamine effect" is rather weakly supported. First, it is unclear how efficient the HisC11-knockdown is. Second, the HisC11 inhibitor cimetidine was used at a concentration that is 20% of that previously documented to be effective (Hong et al. 2006). To be able to make a strong statement the authors should use an existing mutant allele of HisC11 (Hong et al. 2006).

6. Fig. 1I-K suggests a strong reduction in tumor growth upon histidine elimination, but not a complete abrogation of growth (growth to $\approx 140\%$ in the absence of histidine, as opposed to 350% in the presence of histidine). In contrast, Fig. S1C shows 0 tumor growth on a medium lacking histidine ("clonal volume (FC)" is 0 ± 0). Why this discrepancy?

7. In Figs. S4D & H the frequency of "N[act]" tumor clones differs dramatically from that of "scrB Ras[V12]" tumors (which seems to be reflected in the fact that in one case "clone volume" was determined whereas in the other case "GFP/DAPI ratio", i.e. separate clones could presumably not be identified here). This raises two questions: where does this difference come from, and might it affect the biology of the resulting tumors and the ensuing conclusion?

8. The *Drosophila* genotypes should be fully and correctly indicated somewhere. For example, the genotype "w;; FRT82B N[ACT]" is apocryphal and also not stated in the indicated reference. In the absence of such information it is also not clear whether the number of UAS-transgenes is kept constant within one experiment. Thus, e.g. in Fig. 4O part of the effect could be explained by titration of GAL4 leading to reduced expression of relevant transgenes.

Furthermore, while the manuscript is overall well written, some passages need clarification:

9. For the readers' sake the authors should clearly indicate in each experiment whether it was derived from the analysis of larval brains or from adult brains.

10. In general it is not clear whether the sample sizes (e.g. in Fig. 1F) refer to the number of clones or the number of animals from which these clones were analyzed. If the former, $n < 10$ could mean as little as 2 analysed animals - which is too little to draw any reliable conclusion.

11. Some labels in Fig. S3 are cryptic. What does "# Dcp1+ cells/clone vol" mean in Fig. S3E - how is a number of 0.0002 to be interpreted?

The Y-axis in Fig. S3I (labeled "% + vol / clone vol") shows values of ca. 1%. The brain in panel S3G seems to contain a much higher fraction of Myc+ cells within the clone area.

12. The legend to Fig. 4E uses the term "Myc inhibition" in the context of the Myc[P0] allele - this is wrong, it's a hypomorphic Myc allele.

13. The text states "Inhibition of S6K via RNAi", but in the Methods & Figure the authors only describe a dominant-negative form of S6K (S6K[KQ]).

14. The legend to Fig. S3 states "Hdc inhibition did alter cell death...", which is probably not what the authors wanted to say.

Referee #1:

(1) The word "tumour" should be used in a cautious manner. Are N-act expressing clones tumours, or just overgrowing clones? Are nerfin mutant clones in Figure 1-4 tumours? Or just clones of cells that have de-differentiated and are overgrowing? I would try to go through the ms and reduce the tone in this regard and call a clone what is a clone and a tumour what is a tumour. Perhaps a small intro to explain that these clones would give rise to tumours is needed, but what is presented in all figures are clones.

In line with the referee's comments, we have now added an introductory paragraph to explain the relationship between clones and tumours. We have also changed the wording from "tumours" to "clones" throughout the ms. We have shown previously that nerfin-1 mutant clones dedifferentiate, overgrow, and exhibit tumour-like properties, in that they can be transplanted into naïve adult hosts, and induce metastasis (Froldi et al., G&D, 2015).

(2) The first page of the results is a little bit complicated to follow and might need a little bit of re-phrasing for the non-Drosophila and/or non-NB people. The fact that clones are induced in larval development and analysed in adulthood might not very popular among all Drosophilists and might need a couple of sentences to add the corresponding references and explain the logic behind. Second, authors use Leucine as control. Why Leucine? This should be better explain. Third, authors identify a requirement of nerfin-1 mutant clones for Histidine and other EAAs except Valine. Why do then authors focus on His? Again, a small explanation is needed.

We have made the explanatory changes suggested by the reviewer.

(3) In page 7, the wording used to explain the differential impact of addition of histidine relatives to the growth of nerfin clones should be improved. The fact that histamine supplementation is able to rescue the loss of histidine should be better explained. At the end of the paragraph, "showed to be" and not "showed be".

We have made the changes suggested by the reviewer.

(4) In page 9, the first sentence says exactly the opposite as it is been shown. I imagine this is simple wording mistake.

We thank the reviewer for highlighting this wording mistake, which has now been corrected.

(5) The use of the slimfast antisense transgene should be validated with the use of an RNAi transgene.

We have used RNAi against slimfast in addition to Slif antisense transgene. Our data shows that consistent with SlifANTI, Slif RNAi also suppressed nerfin-1 clonal growth, but did not significantly affect the growth of pros clonal growth (Reviewer's Figure 1). However, it has come to our attention during the revision of this manuscript (unpublished data from Hugo Stocker) that Slif RNAi reflects the long antisense transcript going through another AA transporter a couple of genes away (also linked to the TORC1 pathway). Given this uncertainty, we have therefore not included this data in the manuscript, but have presented it as Reviewer's Figure 1. However, we believe omission of this data does not change any of the major conclusions of the manuscript.

Referee #2:

In this work, the authors investigate the effect of diet on the growth of various tumors in *Drosophila melanogaster*. For this purpose, they first develop a chemically defined diet. By individually removing or reducing 9 essential amino acids from this diet, they then identify histidine as being largely dispensable for the development of ovaries or neuroblast clones in the adult, but being limiting for the pathological growth of nerfin-1 mutant NB1- and NB2-derived tumors. A similar dependency on histidine is shown for Notch-induced tumors of the NB2 lineage, but not for prospero-mutant tumors of the NB1 lineage or epithelial tumors induced by mutant Ras. The authors

further provide evidence that the histidine-dependency involves its metabolite histamine, and that it is phenocopied by knockdown of the growth regulators Myc or eIF4E, and overcome by Myc overexpression. Taken together, this study suggests that some tumors (but not all) strongly rely on histidine derivatives for their nucleolar, cellular and overall growth.

The observation that histamine (which to my knowledge cannot be converted to histidine in animal cells) is required for the efficient proliferation of certain tumors is potentially very interesting - it suggests that some function of histidine other than its incorporation in proteins is important for tumor growth. However, the rationale underlying this work is not quite clear to me. In the examples cited in the introduction, certain normally non-essential amino acids (such as glutamine) become essential for some tumors, thereby revealing a (potentially) exploitable vulnerability of these tumors. Here, on the other hand, the authors start out with amino acids that are thought to be essential for all cells and unexpectedly find that histidine is not essential in some situations (they do not mention whether larvae would survive in the complete absence of histidine). To me, this primarily raises the question why normal *Drosophila* neuroblasts and ovaries can cope without histidine. Supposedly animal cells are incapable of synthesizing histidine - so how can these cells keep making proteins to proliferate? If this were a particularity of these *Drosophila* cells, the impact of the present findings would be limited.

We thank the referee for his/her comments. Histidine cannot be synthesized in either humans or Drosophila. We have now clarified the rationale for our work in the revised manuscript by explaining that the EAA deprivation regime is not a total histidine deficiency. The deprivation regime that we use does effectively deplete the internal EAA stores accumulated during development, but it does not affect medium term adult survival or basal protein synthesis. In this study, we were ultimately interested in understanding the differential response of tumours versus normal tissues upon histidine reduction but not total depletion. Our results suggest that nerfin-1 tumours are more sensitive to a reduction of histidine, in comparison to wildtype Drosophila neuroblasts and ovaries. The degree of depletion of EAAs achieved with our dietary withdrawal regime is insufficient to prevent protein synthesis.

Specific criticisms & suggestions:

1. In flies fed a histidine-free diet, the "missing histidine" could be provided by other organs (e.g. by the fat body) or by gut microbes. The latter possibility should be addressed by testing the histidine-dependence in bacteria-free (i.e. antibiotics treated) flies.

The referee is mistaken on this particular point. The NMR experiment in Figure 2B (and S2) shows that whole flies (including their fat body and gut microbes) on CDD-his have levels of histidine under the detection limit (whereas it is detectable in those fed on CDD with histidine). This shows that, on a histidine-free diet, neither the fat body, gut microbes nor any other potential internal source can boost histidine levels to above the detection threshold. In other words, His is missing in flies fed on CDD-His. To aid clarity on this point, we have made it more prominent in the figure legend that the NMR experiments were done on extracts from whole flies.

2. It is not clear whether Myc-overexpression acts tumor cell-autonomously to rescue the growth of "nerfin-1 HDC-knockdown" tumors. In the corresponding experiment the authors use hs-FLP to induce clones throughout the animal - not just in the nervous system. It is well conceivable that Myc-overexpression in some other tissue has a non-autonomous impact on tumor cell growth, as Myc has been shown before to have non-autonomous effects on animal growth. The authors should repeat some of these experiments with the NBII tumor system (driven by "wor-GAL4 ase-GAL80") to reduce such possibly confounding effects.

We thank the reviewer for the suggestion of excluding the possible non-autonomous effect of Myc on nerfin-1 tumour growth. We think that such a non-autonomous effect is unlikely, given that non-CNS clones are randomly induced yet we see CNS clonal phenotypes in all larvae. The suggested experiment with NBII tumour system (with Wor-Gal4 ase Gal80) would not tell us about type I neuroblasts and it is not possible in this instance, as this GAL4 driver does not give us robust phenotypes with the available RNAi lines against Nerfin-1. We know that all the available RNAi lines against nerfin-1 (Trip, VDRC lines) require a driver as strong as tubulin Gal4, as well as UAS-Dicer, to give a robust phenotype (we have done extensive testing of these reagents).

In line with the referee's request, we have used an alternative approach. We have repeated the "Myc-overexpression rescues the growth of "nerfin-1 Hdc-knockdown" experiment" using a MARCM system under the control of neural specific Elav-Gal4, instead of the tub-Gal4 that we used in other parts of this ms. With the Elav-GAL4 system, the UAS transgenes are only expressed in clones in the nervous system (CNS and PNS, Reviewer figure 2). However, while nerfin-1 Hdc1 knockdown using Elav-MARCM showed the same trend as tub-GAL4 MARCM in reducing nerfin-1 clone volume, this difference was not statistically significant, or as dramatic as the effects with tub-MARCM (Reviewer figure 2, compare HdcRi results in E with F). This is likely due to the fact that ElavGal4 is not expressed as strongly as tubGal4. We have also repeated the "Myc-overexpression rescues the growth of "nerfin-1, Hdc-knockdown" experiment with Elav-MARCM but did not see a significant difference either, likely for the same reason. We have included these data as a reviewer's only figure (Reviewer figure 2).

A similar explanation can also be applied to the Myc reduction-of-function experiments, where the whole animal is Myc[P0/+], not just the tumor cells. These latter experiments have the additional caveat that the experimental Myc[P0/+] animals were females, whereas the controls they were compared to [Y/+] were males - and sex has been reported to affect tissue growth in flies, therefore could affect tumor growth here, too.

To overcome the non-autonomous effect of Myc downregulation in nerfin-1 and pros clones, we attempted to knock down Myc with RNAi in nerfin-1 and pros clones using MARCM. However, we found that mycRNAi; pros clones still displayed a very high level of Myc expression, suggesting that mycRNAi is inefficient at knocking down Myc in pros mutant clones (Reviewer Figure 3C-C'). On the other hand, mycRNAi clones in the wing imaginal disc (Reviewer Figure 3A-A') do show an efficient downregulation of Myc. Together these experiments suggested that Myc in pros clones is highly resistant to RNAi mediated downregulation (Reviewer Figure 3), at least with the reagents available to us.

Given the limitation of the experimental reagents, we resorted back to performing the Myc reduction-of-function experiment in whole animals using the myc^{P0} allele. To address the sex specific differences raised by the reviewer, we have repeated our myc^{P0} experiments, using nerfin-1 and pros clones generated in females as a control to directly compare clone size of myc^{P0/+};nerfin-1 and myc^{P0/+};pros clones generated in females, and these new experiments are included in Figure 4. Similar to previous results, we have shown that myc reduction significantly affects nerfin-1 but not pros clonal growth.

3. Myc manipulated flies are characterized in terms of nucleolar, nuclear and tumor size. The observed effects might have been expected, since Myc is known to affect these parameters in many non-transformed cells as well. The authors should determine the composition of the resulting tumors (percentage of NBs, GMCs, differentiated neurons) to determine whether Myc also affects the nature of the tumors.

We thank the reviewer for this suggestion, and have now quantified the clone composition for myc^{P0/+};nerfin-1 vs nerfin-1 and for myc^{P0/+};pros vs pros clones in Figure 4. In wildtype NB clones, NBs are marked by Dpn, Dpn+/Elav+ cells mark GMCs and Dpn-/Elav+ cells mark neurons. In nerfin-1 clones, GMCs as well as semi-dedifferentiated neurons (See Figure 2F schematic) are both Dpn+/Elav+, therefore, it is not possible to distinguish between these two populations. Therefore, we have limited our clone composition analyses to the proportion of Dpn+ (NBs) and Elav+(neurons) per clone (Figure 4I and N)

4. It is not clear to which extent the various regimes of larval clone induction affect the duration of development until adult eclosion - it is not stated in the Methods section that this duration was controlled. Alterations could of course affect tumor size and make it hard / impossible to compare the different experiments.

We have now added the details of larval clone induction regime to the relevant experimental figure legends. Larval clone induction of 8 to 15 minutes for tubGal4 MARCM experiments does not affect the timing of larval development until adult eclosion. In addition, the same heat shock regime is applied within each experimental group.

5. The statement that the histamine receptor HisC11 is not involved in the observed "histamine effect" is rather weakly supported. First, it is unclear how efficient the HisC11-knockdown is. Second, the HisC11 inhibitor cimetidine was used at a concentration that is 20% of that previously documented to be effective (Hong et al. 2006). To be able to make a strong statement the authors should use an existing mutant allele of HisC11 (Hong et al. 2006).

We thank the reviewer for this suggestion. However, it is not technically feasible to make double mutant MARCM clones of HisC11 and nerfin-1 as they are on different chromosome arms. We have instead strengthened the evidence that HisC11 is not involved by performing two additional experiments: 1) we fed cimetidine at 500mg/ml as demonstrated to be effective at altering Drosophila temperature preference (phenocopying the effects of hdc mutant) in Hong et al., 2006 (Figure EV3A). 2) we performed HisC11 knock down using an RNAi line that has previously been shown to effectively knockdown HisC11 (VDRC 104966, Oh, 2013) and we found HisC11 inhibition did not significantly alter the growth of nerfin-1 clones (Figure EV3B).

6. Fig. 11-K suggests a strong reduction in tumor growth upon histidine elimination, but not a complete abrogation of growth (growth to $\approx 140\%$ in the absence of histidine, as opposed to 350% in the presence of histidine). In contrast, Fig. S1C shows 0 tumor growth on a medium lacking histidine ("clonal volume (FC)" is 0 +/- 0). Why this discrepancy?

In Fig S1C we measured the total volume of nerfin-1 clones per CNS. In contrast, in Figure 11-K each data point represented the volume of individual nerfin-1 clones. The discrepancy is therefore likely due to the difference in the quantification methodology. We have clarified the difference in methodology, both in the methods section as well as in the figure legends.

7. In Figs. S4D & H the frequency of "N[act]" tumor clones differs dramatically from that of "Scrib Ras[V12]" tumors (which seems to be reflected in the fact that in one case "clone volume" was determined whereas in the other case "GFP/DAPI ratio", i.e. separate clones could presumably not be identified here). This raises two questions: where does this difference come from, and might it affect the biology of the resulting tumors and the ensuing conclusion?

In S4D-F, while E and F are single confocal sections, we showed a maximum projection image for D. Thus, it appeared that the frequency of clone induction differed between N[ACT] tumour and Scrib Ras[V12] tumours. In fact, there is no difference in the heat shock regime, and no significant difference in the frequency of the clones between the two experiments. We thank the reviewer for picking up this and we have now replaced the max projection with a single confocal section of N[ACT] in Figure EV4D. We have also quantified clone volumes of discrete clones for N[ACT] (instead of GFP/DAPI ratio), and this data is now Figure EV4G.

8. The Drosophila genotypes should be fully and correctly indicated somewhere. For example, the genotype "w;; FRT82B N[ACT]" is apocryphal and also not stated in the indicated reference.

We have corrected this in the methods section.

In the absence of such information it is also not clear whether the number of UAS-transgenes is kept constant within one experiment. Thus, e.g. in Fig. 4O part of the effect could be explained by titration of GAL4 leading to reduced expression of relevant transgenes.

We have removed the original Fig 4O from the manuscript. The number of transgenes is now kept constant in all experiments.

Furthermore, while the manuscript is overall well written, some passages need clarification:

9. For the readers' sake the authors should clearly indicate in each experiment whether it was derived from the analysis of larval brains or from adult brains.

We have added this information throughout the ms and to the figure legends.

10. In general it is not clear whether the sample sizes (e.g. in Fig. 1F) refer to the number of clones

or the number of animals from which these clones were analyzed. If the former, $n < 10$ could mean as little as 2 analysed animals - which is too little to draw any reliable conclusion.

In all experiments throughout the ms, the sample size refers to the number of clones. Figure 1F was the exception, where the graph represents the total tumour volume in each CNS. We have added this information to the methods and the figure legend.

11. Some labels in Fig. S3 are cryptic. What does "# Dcp1+ cells/clone vol" mean in Fig. S3E - how is a number of 0.0002 to be interpreted?

We have replaced Figure EV3E with a graph showing the absolute number of Dcp1+ cells per clone.

The Y-axis in Fig. S3I (labeled "% + vol / clone vol") shows values of ca. 1%. The brain in panel S3G seems to contain a much higher fraction of Myc+ cells within the clone area.

We have corrected the Y-axis labelling in Figure EV3I.

12. The legend to Fig. 4E uses the term "Myc inhibition" in the context of the Myc[P0] allele - this is wrong, it's a hypomorphic Myc allele.

We have corrected this in the figure legend.

13. The text states "Inhibition of S6K via RNAi", but in the Methods & Figure the authors only describe a dominant-negative form of S6K (S6K[KQ]).

We have corrected the mistake in the text, we have only used the dominant-negative transgene for S6K in our studies.

14. The legend to Fig. S3 states "Hdc inhibition did alter cell death...", which is probably not what the authors wanted to say.

We have now corrected this in the text.

2nd Editorial Decision

26th Nov 2018

Thank you for submitting your revised manuscript to The EMBO Journal. Your revision has now been seen by the original referees.

Both referees appreciate that the introduced revisions have strengthened the findings. However, referee #2 also has some remaining points about the data and the revisions. I have also asked referee #1 about these points and the referee is in agreement that the points raised are valid and should be addressed. Some of the points raised can be addressed with a better explanation of how the experiments were done. Let me know if we need to discuss anything in more details.

REFeree REPORTS:

Referee #1:

Authors have addressed most of my concerns. Again, the term "tumour" has to be used in an extremely cautious manner if we, Drosophilists, do not want to bother the mammalian community. This word is still being used throughout the ms instead of "clone". Larvae do not bear "tumors", they do contain "clones". Please, correct the word in all cases.

I have gone throughout referee 2's comments and I agree that authors should carefully address all concerns raised by that reviewer. In general, authors should properly solve all discrepancies, many of them as a consequence of bad selection of the illustrative example, bad interpretation of the

results or poor clarification of the quantification being made in the figures. In all cases, this is a sign of sloppiness, and should be properly corrected.

Referee #2:

The authors have addressed some of the reviewers' criticisms. However, several issues remain, including questions about data quality/interpretation, that prevent me from recommending publication at this time.

1. I have not been able to find Reviewers' Figures 1-3 and hence was unable to judge the pertaining points.
2. I have previously been puzzled by the difference between Fig. 11-K (where after 9d of adult growth nerfin-1 clones on "-his" reach about 40% the size of nerfin-1 clones on "CDD") and Fig. S1C (where the volume of nerfin-1 clones on "-his" amounts to 0% of that on "CDD"). The authors' rebuttal has done nothing to alleviate this confusion.
3. On p. 6 of the Results section the authors seem to imply that the NMR detection limit for histidine is 0.01 mM, hence that histidine-starved adults contain <0.01 mM histidine. It is unclear where this value of 0.01 mM comes from. The only NMR data is found in FigEV 2A: although the meaning on the left y-axis is not explained ("+", "2x", "-"), I assume that the "+" lanes correspond to flies fed with histidine-replete medium. If this his peak reflects a concentration of 2 mM (Fig. 2B), I don't believe that 0.01 mM (i.e. 1/200 of this concentration) are still reliably detectable.
4. The representative "myc/+, pros" clone in Fig.4F is obviously clearly smaller than the representative "+/+, pros" clone in Fig.4E. However, the quantitation in Fig. 4H suggests that that there is no size difference between the two genotypes. This does not fit together.
5. The y-axis label of FigEV 3I "% + vol/clone" suggests that 100% of nerfin-1 clones are "Myc-positive" and "Myc+ELAV-positive". However, the nerfin-1 clone in FigEV 3G shows strongly reduced ELAV-staining intensity as compared to the surrounding tissue, and regions of the clone appear to be negative for either Myc or ELAV. This does not fit with the quantitative analysis of the clones.
Also, it is hard to believe that the "nerfin-1 HDC-RI" clone in Fig EV3H has only 40% as much Myc-staining as the "nerfin-1" clone in FigEV 3H (as suggested by the bar graph in FigEV 3I).
6. The authors state on p. 9 of the Results "We observed that the growth of eye NACT clones, despite a dependency on Myc (Figure EV4D-G), was not significantly decreased by Hdc knockdown." This statement is misleading, to say the least. The size effect of HDC-KD may not reach significance, whereas it does for Myc-KD (although I'm pretty sure that the difference in p-values between the two settings is minor) - but FigEV 4G shows that the overall effects of HDC-KD and Myc-KD on clonal size are virtually the same.
7. (minor point) what does the y-axis title "Elav+ volume (FC)" in Fig 3C stand for? in other figures "FC" indicates "fold change", but then it doesn't make sense to set "fold change" for "CDD" control at 0.4.

2nd Revision - authors' response

7th Dec 2018

Referee #1:

Authors have addressed most of my concerns. Again, the term "tumour" has to be used in an extremely cautious manner if we, Drosophilists, do not want to bother the mammalian community. This word is still being used throughout the ms instead of "clone". Larvae do not bear "tumors", they do contain "clones". Please, correct the word in all cases.

I have gone throughout referee 2's comments and I agree that authors should carefully address all concerns raised by that reviewer. In general, authors should properly solve all discrepancies, many of them as a consequence of bad selection of the illustrative example, bad interpretation of the

results or poor clarification of the quantification being made in the figures. In all cases, this is a sign of sloppiness, and should be properly corrected.

We have now replaced all wordings in the ms referring to 'tumours' by 'clones'.

Referee #2:

The authors have addressed some of the reviewers' criticisms. However, several issues remain, including questions about data quality/interpretation, that prevent me from recommending publication at this time.

1. I have not been able to find Reviewers' Figures 1-3 and hence was unable to judge the pertaining points.

We have now attached the reviewer's figures 1-3 below.

Reviewer's Figure 1: RNAi against Slif differentially affect *pros* and *nerfin-1* clonal growth
pros and *nerfin-1* clonal growth depends differentially on amino acid transporter Slif, inhibition mediated via RNAi (VDRC 101643). (*nerfin-1*, n= 54, 56, *pros*, n=22,26)

Reviewer's Figure 2: Neural specific ElavGal4 MARCM is considerably weaker than tubGal4 MARCM in driving transgene expression

(A-D) [$P\{w[+] \text{ elavGAL4}[c155]\}$, $P\{ry[+] \text{ hsFLP}1\}$; $CyO / P\{w[+] \text{ UAS-nucZ}\}20b$, $P\{w[+] \text{ UAS-CD8:GFP}\} LL5$; $TM6, Tb, Hu / P\{w[+] \text{ tubP-GAL80}\} L9$, $P\{w[+] \text{ FRT } 2A\}$] or ElavGal4 MARCM generates GFP labelled clones only in the CNS and the disc, and not in the fatbody or the gut. (E), we observed ~60% reduction in *nerfin-1* clone volume upon knocking down of RNAi Hdc with tub-Gal4 MARCM. (F), in contrast, we only observed ~30% reduction ($p > 0.05$) in *nerfin-1* clone volume upon knocking down of Hdc using elav-Gal4 MARCM. In addition, overexpression of Myc did not significantly alter *nerfin-1* clone volume. (G) clone composition of *nerfin-1* was not significantly altered by overexpression of Myc, and knockdown of Hdc via RNAi. (H) Overexpression of Myc did not significantly alter clone size or (I) clone composition of *nerfin-1;HdcRNAi* clones

Reviewer Figure 3: MycRNAi can decrease Myc expression in wing disc but not *pros*¹⁷ CNS clones.

Using tubGal4 MARCM, we overexpressed Myc RNAi in *pros* clones, we found that Myc levels were down regulated in the wing imaginal disc clones (A-A'), in CNS clones, Myc expression was still upregulated despite Myc knockdown via RNAi.

2. I have previously been puzzled by the difference between Fig. 1I-K (where after 9d of adult growth nerfin-1 clones on "-his" reach about 40% the size of nerfin-1 clones on "CDD") and Fig. S1C (where the volume of nerfin-1 clones on "-his" amounts to 0% of that on "CDD"). The authors' rebuttal has done nothing to alleviate this confusion.

As seen below (screen shots of velocity quantification), the panels on the left are representative CNSs from the experiment quantified in Figure EV1C. Due to the long heat shock (1hr), frequent clones were generated, and on CDD, the overgrowing clones merge together to form a single large GFP+ object. For unknown reasons, clones were more frequently generated on CDD compared to -His, where the clones were significantly smaller and less frequent. Never the less, as we could not quantify the size of individual clones on CDD, we plotted total GFP+ volume per CNS in our quantification. Therefore, in Figure EV1, nerfin-1 tumours are around 100-fold larger on CDD compared to -His, raw data from these quantifications is attached below.

We then looked at this more carefully in Figure 1K (right panel), where heat shock was performed 24hrs later (in order to shorten the period of time between tumour induction and dissection), and animals were heat shocked for only 10 minutes, to ensure that clones were generated infrequently and well separated. As you can see in these images, CDD clones while still significantly larger than that of -His, the difference between individual clone volume was less striking. Again the raw data from Figure 1K is plotted below.

Therefore, the differences in fold change between Figures EV1 and Figure 1 can be accounted for by the difference in the heat shock regimes, where the total clone volume per CNS was plotted in Figure EV1, and discrete clone volume was plotted in Figure 1.

3. On p. 6 of the Results section the authors seem to imply that the NMR detection limit for histidine is 0.01 mM, hence that histidine-starved adults contain <0.01 mM histidine. It is unclear where this value of 0.01 mM comes from. The only NMR data is found in FigEV 2A: although the meaning on the left y-axis is not explained ("+", "2x", "-"), I assume that the "+" lanes correspond to flies fed with histidine-replete medium. If this his peak reflects a concentration of 2 mM (Fig. 2B), I don't believe that 0.01 mM (i.e. 1/200 of this concentration) are still reliably detectable.

The value of 0.01 mM was based on the directly measured concentration in the NMR tube (assuming a fixed concentration of the DSS standard) and not the VDTS back-calculated value for the fly itself. We have therefore calculated a more biologically meaningful value for the fly itself, yielding a conservative, VDTS-derived value for the limit of detection for histidine as ~ 0.3 mM. The main text has been adjusted accordingly. We acknowledge the reviewer's comments regarding the lack of clarity for the key and legend for FigEV 2. The legend has been revised accordingly to read:

Figure S2 related to Figure 2: ^1H NMR can detect changes in dietary histidine and his dietary manipulation differentially affects nerfin-1- and pros- tumour growth

A) spectra showing a region from 8.0 -7.1 ppm from extracted whole adult fly polar metabolome before (**upper panel**) and after (**lower panel**) clearance of dietary histidine from the gut (see methods), boxed panel highlights the histidine peak. Replica peaks for histidine are seen in the boxed areas at ~ 7.9 (ϵ -proton) and 7.1 (δ -proton) ppm in histidine-replete: "+" (0.5 g/L his) and "2x" (1.0 g/L his) CDD profiles, histidine peaks are not seen in "-" (0 g/L his) profiles in either panel.

B) nerfin-1 increased tumour volume by ~ 4 fold upon feeding on 2x his ($n=24,15$)

C) pros⁺ tumour volume was not significantly changed upon feeding on 2x his ($n=19,34$)

4. The representative "myc/+, pros" clone in Fig.4F is obviously clearly smaller than the representative "+/+, pros" clone in Fig.4E. However, the quantitation in Fig. 4H suggests that that there is no size difference between the two genotypes. This does not fit together.

We have replaced Figure 4 E-F with images more representative of our quantifications of in 4H.

5. The y-axis label of FigEV 3I "% + vol/clone" suggests that 100% of nerfin-1 clones are "Myc-positive" and "Myc+ELAV-positive". However, the nerfin-1 clone in FigEV 3G shows strongly reduced ELAV-staining intensity as compared to the surrounding tissue, and regions of the clone appear to be negative for either Myc or ELAV. This does not fit with the quantitative analysis of the clones.

The original data was normalised to nerfin-1, and we agree with the reviewer that the y axis label did not make sense. The data is now replotted as "% +vol/clone".

Also, it is hard to believe that the "nerfin-1 HDC-RI" clone in Fig EV3H has only 40% as much Myc-staining as the "nerfin-1" clone in FigEV 3H (as suggested by the bar graph in FigEV 3I).

We have replaced Figure EV3G-H with images more representative of our quantifications in Figure EV3I. We have also added additional examples of representative images of Myc⁺ cells per clone in nerfin-1 vs. HdcRi;nerfin-1 clones below, to illustrate that there is a significant and reproducible reduction in Myc⁺ cells in nerfin-1 clones upon Hdc knockdown, consistent with our quantifications in EV3I.

6. The authors state on p. 9 of the Results "We observed that the growth of eye NACT clones, despite a dependency on Myc (Figure EV4D-G), was not significantly decreased by Hdc knockdown." This statement is misleading, to say the least. The size effect of HDC-KD may not reach significance, whereas it does for Myc-KD (although I'm pretty sure that the difference in p-values between the two settings is minor) - but FigEV 4G shows that the overall effects of HDC-KD and Myc-KD on clonal size are virtually the same.

We have re-examined the data, and found that the distribution of the data points in Figure EV4G does not fit a normal distribution (as seen in box-plot below). Using non-parametric t-test, we found Hdc knockdown did significantly reduce NACT clone size. We have now corrected this in both Figure EV4G and the corresponding text in the manuscript, which now reads: "We observed that the growth of N^{ACT} clones in the eye discs which exhibited dependency on Myc was also significantly decreased by Hdc knockdown (Figure EV4D-G)."

7. (minor point) what does the y-axis title "Elav+ volume (FC)" inf Fig 3C stand for? in other figures "FC" indicates "fold change", but then it doesn't make sense to set "fold change" for "CDD" control at 0.4.

This mistake has been corrected, and the Y axis has been changed to "%Elav⁺ volume/clone".

3rd Editorial Decision

9th Jan 2019

Thank you for submitting your revised manuscript to The EMBO Journal. Your study has now been re-reviewed by the referees and their comments are provided below.

As you can see from the comments below, referee #2 still has some hesitations with the analysis. I have taken a careful look at the issues raised and I think you have done a good job in responding to the concerns raised. Referee #1 is of similar opinion. Given this I am very pleased to let you know that we can accept the manuscript for publication here. You can respond to the concerns raised by referee #2 in the point-by-point response or if you wish in the text.

I have asked our publisher to do their pre-publication check on the manuscript and they have made some comments in the figure legend. Please check the word document called Wiley pre-acceptance check and please incorporate their suggestions.

Once we get the revised version in I will send you the formal acceptance letter.

Congratulations on a nice paper

 REFEREE REPORTS:

Referee #1:

As Reviewer 1 of this ms, I have gone through Reviewer 2's comments and the point by point response of authors to these comments. I believe authors have addressed all concerns in a satisfactory manner by including more illustrative images and changing the way graphs were represented in some figures in the ms. Authors have also well argued why the differences in the fold changes of experiment vs control in two independent figures (point nr 2) and have included data in three new specific figures to address point 1.

Concerning my minor concern (the word tumour), authors have addressed it properly.

I strongly support publication of this ms in EMBO Journal the way it is, as authors have addressed all major concerns.

Referee #2:

I have mixed feelings about this manuscript. On one hand, it describes an interesting observation that should be published (somewhere). On the other hand, my misgivings have not been reduced by

the two revisions - mainly because of two points.

The first concerns the core message of the manuscript:

the authors' response to point 6 has made it clear to me why Fig. 1I-K and Fig. S1C report (quantitatively) different consequences of histidine withdrawal on nerfin-1-mutant clone growth. However, this explanation indicates that a major effect of histidine withdrawal is a reduction in clone number. This suggests that apoptosis plays a major role upon histidine withdrawal: it is conceivable that the smaller clones (induced at 72h ALH) are eliminated more easily (once histidine is withdrawal from the adults) than the clones that were induced at 48h ALH and therefore have reached a bigger size at the moment they have to face a histidine-less diet. This in turn suggests that at least some of the "growth" effects that are analyzed in the present work are caused by differential effects on apoptosis and cell survival - and it begs for some additional experiments.

The second point has to do with the quality of the data. Several of the reviewer criticisms led to significant changes in the data:

Point 3: the NMR-detection limit for histidine was corrected from 0.01 mM to 0.3 mM (note however, that Fig. 2B still conveys the impression that histidine levels in flies drop from 2 mM in CDD to 0 mM in "- histidine", as there is no indication of the detection limit in the Figure itself - this has to be considered as quite misleading).

Point 5: the axis labeling in Fig. S3I was changed, so that the former "100%" now corresponds to ca. 17% (column 1), viz. 13% (column 3). Along the way, the relative heights of the bars and relative lengths of the error bars have also changed.

Point 6: the significance level for "N[act] HDC[RI]" in Fig. S4C was recalculated and now found to pass the " $p < 0.05$ " cutoff (which it hadn't before).

In addition, two figure panels were exchanged so as to show more "representative examples" (Point 4 & 5).

For all of these changes the authors provide convincing explanations. Nevertheless, having to make such corrections (and several more in the first round of revisions) indicates substantial sloppiness in preparing the manuscript in the first place - and it makes me suspect that additional errors might be hidden in this work that neither of the reviewers happened to catch.

3rd Revision - authors' response

9th Jan 2019

The authors' response to point 6 has made it clear to me why Fig. 1I-K and Fig. S1C report (quantitatively) different consequences of histidine withdrawal on nerfin-1-mutant clone growth. However, this explanation indicates that a major effect of histidine withdrawal is a reduction in clone number. This suggests that apoptosis plays a major role upon histidine withdrawal: it is conceivable that the smaller clones (induced at 72h ALH) are eliminated more easily (once histidine is withdrawal from the adults) than the clones that were induced at 48h ALH and therefore have reached a bigger size at the moment they have to face a histidine-less diet. This in turn suggests that at least some of the "growth" effects that are analyzed in the present work are caused by differential effects on apoptosis and cell survival - and it begs for some additional experiments.

We respectfully disagree with the reviewer's comments regarding 'the histidine withdrawal primarily reduces clone number'. We have not observed significant differences in cell death rate within and outside of CDD vs -His or 25% His clones (Fig 3C and Fig EV 3E and data not shown) to support the hypothesis that -his diet induces elimination of clones.

Corresponding Author Name: Louise Cheng

Journal Submitted to: Karin Dumstreil

Manuscript Number: EMBOJ-2018-99895